



**Assessing improvements in global ocean $pCO_2$ machine learning reconstructions with**
**Southern Ocean autonomous sampling**
Thea H. Heimdal[1], Galen A. McKinley[1], Adrienne J. Sutton[2], Amanda R. Fay[1], Lucas Gloege[3]
[1]Columbia University and Lamont-Doherty Earth Observatory, Palisades, NY, USA
[2]Pacific Marine Environmental Laboratory, National Oceanic and Atmospheric Administration,
Seattle, WA, USA
[3]Open Earth Foundation, Marina del Rey, CA, USA
*Correspondence to:* Thea H. Heimdal (theimdal@ldeo.columbia.edu)
**Abstract**
The Southern Ocean plays an important role in the exchange of carbon between the atmosphere
and oceans, and is a critical region for the ocean uptake of anthropogenic $CO_2$. However, estimates
of the Southern Ocean air-sea $CO_2$ flux are highly uncertain due to limited data coverage. Increased
sampling in winter and across meridional gradients in the Southern Ocean may improve machine
learning (ML) reconstructions of global surface ocean $pCO_2$. Here, we use a Large Ensemble
Testbed (LET) of Earth System Models and the $pCO_2$-Residual reconstruction method to assess
improvements in $pCO_2$ reconstruction fidelity that could be achieved with additional autonomous
sampling in the Southern Ocean added to existing Surface Ocean $CO_2$ Atlas (SOCAT)
observations. The LET allows us to robustly evaluate the skill of $pCO_2$ reconstructions in space
and time through comparison to 'model truth'. With only SOCAT sampling, Southern Ocean and
global $pCO_2$ are overestimated, and thus the ocean carbon sink is underestimated. Incorporating
Uncrewed Surface Vehicle (USV) sampling increases the spatial and seasonal coverage of
observations within the Southern Ocean, leading to a decrease in the overestimation of $pCO_2$. A
modest number of additional observations in southern hemisphere winter and across meridional
gradients in the Southern Ocean leads to improvement in reconstruction bias and root-mean
squared error (RMSE) can be improved by as much as 65 % and 19 %, respectively, as compared
to using SOCAT sampling alone. Lastly, the large decadal variability of air-sea $CO_2$ fluxes shown
by SOCAT-only sampling, may be partially attributable to undersampling of the Southern Ocean.



## 1. Introduction

The ocean plays an important role in mitigating against climate change by sequestering anthropogenic carbon emissions. Since 1850, the oceans have removed a total of $170 \pm 35$ Gt of carbon (Friedlingstein et al., 2022). In order to fully understand the climate impacts from rising emissions, it is essential to accurately quantify the air-sea $CO_2$ flux and the global ocean carbon sink in space and time. The Surface Ocean $CO_2$ ATlas (SOCAT; Bakker et al., 2016) is the largest global database of surface ocean $CO_2$. It contains over 33 million high-quality direct shipboard measurements of $fCO_2$ (uncertainty of $< 5$ µatm), which have been gathered since 1957 (Bakker et al., 2022). However, due to limited resources for ocean observing, limited number of ships/routes, inaccessible regions and unsafe waters, the database covers only about 1% of the global ocean at monthly $1° \times 1°$ spatial resolution over the period of 1982-2023, and is highly biased towards the northern hemisphere.

Observation-based data products have been developed to better constrain surface ocean $pCO_2$ in space and time by extrapolating to global coverage from the sparse SOCAT observations (e.g., Landschützer et al., 2014; Rödenbeck et al., 2015; Gloege et al., 2022; Bennington et al., 2022a,b). These data products utilize machine learning (ML) algorithms to estimate a non-linear function between a suite of driver variables (i.e., sea surface temperature; SST, sea surface salinity; SSS, mixed layer depth; MLD, Chlorophyll; Chl-a, $xCO_2$; atmospheric $CO_2$) and ocean $pCO_2$ (the target variable) where these are co-located. The driver variables are proxies for processes influencing ocean $pCO_2$. Full-coverage driver variable datasets are then processed through these ML algorithms to produce estimated global full-coverage surface ocean $pCO_2$. Since the data products rely on observations to train the algorithms and thus produce these relationships, data sparsity remains a fundamental limitation to this technique.

It has been suggested that targeted sampling from autonomous platforms combined with ships, filling in the state space of $pCO_2$, represent a likely path forward to improve surface ocean $pCO_2$ reconstructions (Bushinsky et al., 2019; Gregor et al., 2019; Gloege et al., 2021; Djeutchouang et al., 2022; Landschützer et al., 2023). One major obstacle, however, is that the indirect $pCO_2$ estimates from floats have high uncertainties ($\pm 11.4$ µatm) and may be biased by as much as $\sim 4$ µatm (Bakker et al., 2016; Williams et al., 2017; Fay et al., 2018; Gray et al., 2018; Sutton et al., 2021; Mackay and Watson 2021; Wu et al 2022). Biases and uncertainties can have



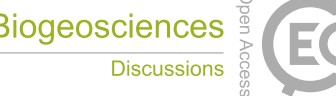

large impacts on global air-sea $CO_2$ flux estimates, given that the global mean air-sea
disequilibrium is only 5-8 µatm (McKinley et al., 2020). It is therefore critical that bias and
uncertainty corrections are well-constrained over different oceanic conditions and over time.

63          Uncrewed Surface Vehicles (USVs), such as those manufactured and maintained by

Saildrone Inc., represent a new type of autonomous platform that can obtain direct $pCO_2$
observations with significantly lower uncertainties compared to other autonomous methods, and
equivalent to the highest-quality shipboard measurements contained in SOCAT (± 2 µatm; Sabine
et al., 2020; Sutton et al., 2021). Such improvements in sampling are critically important in the
undersampled Southern Ocean. This region is fundamental in terms of the ocean's ability to
remove carbon from the atmosphere, being responsible for ~ 40% of the global ocean uptake of
anthropogenic $CO_2$ (Khatiwala et al., 2009). Improved data coverage in the Southern Ocean
represents thus a major opportunity to advance our understanding of the global ocean carbon sink
(Lenton et al., 2006, 2013; Takahashi et al., 2009; Monteiro et al., 2015; Gregor et al., 2019; Gray
et al., 2018; Mongwe et al., 2018; Bushinsky et al., 2019; Sutton et al., 2021; Long et al., 2021;
Mackay et al., 2022; Wu et al., 2022; Landschützer et al., 2023). A combination of SOCAT and
Saildrone USV observations would include high accuracy data from both the long record and
global coverage of ship tracks, and the expanded finer resolution of spatial and seasonal coverage
of the poorly sampled Southern Ocean. Importantly, Saildrone USVs are also able to cover the
spatial extent and seasonal cycle of the meridional gradients, which has been shown to be critical
in order to reduce errors in reconstructing surface ocean $pCO_2$ (Djeutchouang et al., 2022). A
combined approach, with autonomous samples such as those obtained from Saildrone USVs, in
addition to high-quality observations collected from ships, represents thus a promising solution to
improve surface ocean $pCO_2$ ML reconstructions.

83          Here, we assess to what extent surface ocean $pCO_2$ reconstructions can improve by

implementing the $pCO_2$-Residual machine learning (ML) reconstruction (Bennington et al., 2022a)
with the combined inputs of SOCAT and Saildrone USV coverage. However, instead of using
actual observations, we sample the target (i.e., surface ocean $pCO_2$) and driver variables (i.e., SST,
SSS, MLD, Chl-a and $xCO_2$) from our Large Ensemble Testbed (LET) of Earth System Models
(ESMs) (e.g., Stamell et al., 2020; Gloege et al., 2021; Bennington et al., 2022a). There are two
major benefits of using a testbed compared to actual observations. First, in an ESM, surface ocean





pCO$_2$ is known at all times and locations. Therefore, the pCO$_2$ reconstructed by the ML algorithm
can be robustly evaluated in space and time against a known 'truth' (i.e., 'model truth'). The
reconstruction evaluation is thus not limited to the availability of sparse real-world ocean
observations. Secondly, a testbed can be used to plan and evaluate the impact of different sampling
strategies on the reconstructed pCO$_2$. It is important to stress that, by using a model testbed, we do
not predict real-world surface ocean pCO$_2$ and air-sea CO$_2$ fluxes. The goal here is to assess the
accuracy with which an ML algorithm can reconstruct the 'model truth' given inputs of samples
consistent with real-world data coverage from the SOCAT database and Saildrone USVs.

98       By utilizing the observational coverage of SOCAT and Saildrone USV transects, we assess

to what extent the pCO$_2$-Residual method accurately reconstructs model surface ocean pCO$_2$ in
space and time. Additionally, we explore the timing, magnitude, duration and spatial extent of
Southern Ocean USV sample additions that most significantly improve the pCO$_2$ predictions.
**2. Methods**
*2.1 The Large Ensemble Testbed (LET)*
In this study, the Large Ensemble Testbed (LET) includes 25 members from three independent
initial-condition ensemble models (i.e., CanESM2, CESM-LENS and GFDL-ESM2M; Kay et al.,
2015; Rodgers et al., 2015; Fyfe et al., 2017), giving a total of 75 members within the testbed. We
do not use the MPI-GE model that was included in the past LET studies because its Southern
Ocean pCO$_2$ seasonality and decadal variability appears to be anomalously large (Gloege et al.,
2021; Fay and McKinley, 2021; Bennington et al., 2022a). Each individual Earth System Model
(ESM) is an imperfect representation of the actual Earth system, so the multiple Large Ensembles
are used to span different model structures and their representation of internal variability. Each
ensemble member undergoes the same external forcing (i.e., historical atmospheric CO$_2$ before
2005 and Representative Concentration Pathway 8.5 through 2016, plus solar and volcanic
forcing), but the spread across the ensemble members gives a unique trajectory of the ocean-
atmosphere state over time, i.e., a different state of internal variability as well as the difference
across models.
The LET used in this study includes monthly 1°x1° model output from 1982-2016 (Gloege
et al., 2021). For each individual ensemble member of the LET, surface ocean pCO$_2$ and co-located





driver variables (i.e., SST, SSS, Chl-a, MLD, $xCO_2$) were sampled monthly at a 1°x1° resolution,
at times and locations equivalent to SOCAT and Saildrone USV observations (**Fig. 1**; Step 1).
While the SOCAT observations were sampled from the testbed matching the actual years of
sampling, the USV observations were sampled from the testbed starting in year 2007 (for ten-year
sampling) or 2012 (for five-year sampling) (see **Sect. 2.4**). As our focus is on reconstruction for
the open ocean, testbed output for coastal areas, the Arctic Ocean (>79°N) and marginal seas
(Hudson Bay, Caspian Sea, Black Sea, Mediterranean Sea, Baltic Sea, Java Sea, Red Sea and Sea
of Okhotsk) were removed prior to algorithm processing.

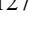

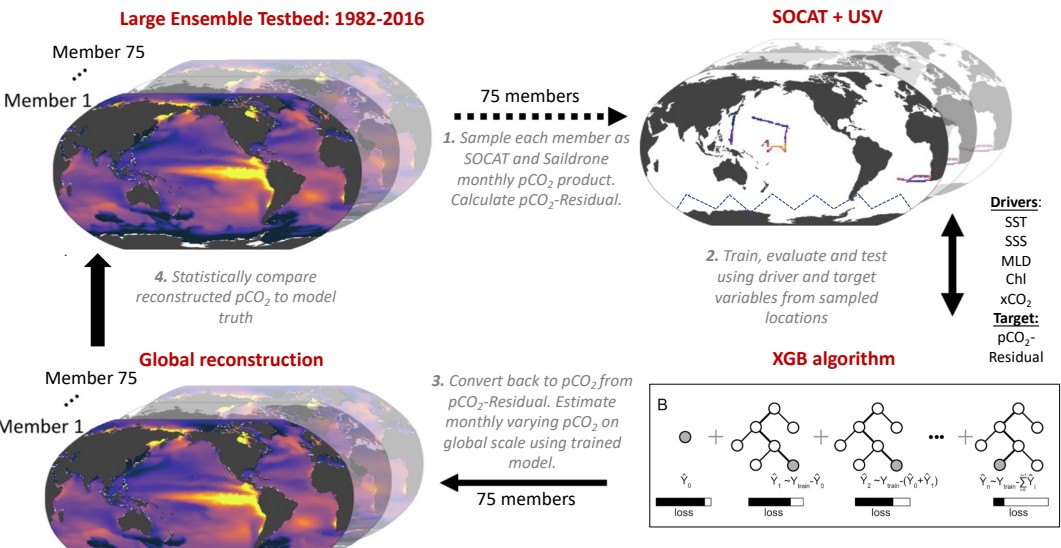

**Figure 1:** Schematic of the Large Ensemble Testbed (LET; modified from Gloege et al., 2021). **1:** Surface ocean
$pCO_2$ from each of the 75 model members is sampled in space and time mimicking real-world SOCAT and Saildrone
USV observations (see **Fig. 2**; **Table 1**; **Section 2.5**). Prior to algorithm processing, $pCO_2$-Residual is calculated, i.e.,
the direct effect of temperature has been removed from the $pCO_2$ value (**Section 2.2**). **2:** The $pCO_2$-Residual (target
variable) and co-located driver variables (i.e., SST, SSS, MLD, Chl, $xCO_2$) sampled from the testbed are processed
by the XGBoost (XGB) algorithm (**Section 2.3**). **3:** Based on the full-coverage of driver variables, $pCO_2$-Residual is
reconstructed globally. This process is repeated 75 times, individually for every single testbed model member. The
temperature component ($pCO_2$-T) is then added back to the $pCO_2$-Residual for each value. **4:** Since we are using
model testbed and not real-world observations, the globally reconstructed $pCO_2$ can be evaluated against the 'model
truth' at all 1°x1° grid cells, not just where observations are available. SST = sea surface temperature. SSS = sea
surface salinity. MLD = mixed layer depth. Chl = chlorophyll. $xCO_2$ = atmospheric concentration of $CO_2$.






*2.2 The pCO₂-Residual approach*
We used the pCO$_2$-Residual approach following Bennington et al. (2022a), which removes the
well-studied direct effect of temperature on pCO$_2$ from the LET model output prior to algorithm
processing. Temperature has both direct and indirect effects on surface ocean pCO$_2$. The direct
effect of temperature, due to solubility and chemical equilibrium, is that an increase in temperature
directly causes an increase in pCO$_2$ (Takahashi et al., 1993). Indirectly, temperature changes are
associated with biological production and wintertime vertical mixing; and these processes tend to
result in opposing pCO$_2$ changes. To build reconstruction algorithms through the data-driven
training that occurs in ML, the statistics in all other algorithms developed to date must identify a
function that disentangles these competing effects of SST on pCO$_2$. Here, the algorithm is assisted
by removing this known temperature effect, and it must therefore only learn the pCO$_2$ impacts
from biogeochemical drivers. The pCO$_2$-Residual method leads to physically understandable
connections between the input data and output (Bennington et al., 2022a), which mitigates to some
degree 'black box' concerns typically associated with ML algorithms (Toms et al., 2020). Further,
this method has been shown to perform better against independent observations than other
common observation-based products (Bennington et al., 2022a). A brief description is provided
here, but for further details see Bennington et al. (2022a).
The temperature-driven component of pCO$_2$ (pCO$_2$-T) is calculated using this equation:
$$\text{pCO}_2\text{-T} = \text{pCO}_2{}^{\text{mean}} * \exp[0.0423 * (\text{SST} - \text{SST}^{\text{mean}})]$$
where pCO$_2{}^{\text{mean}}$ and SST$^{\text{mean}}$ is the long-term mean of surface ocean pCO$_2$ and temperature,
respectively, using all 1°x1° grid cells from the testbed. Once pCO$_2$-T is determined, pCO$_2$-
Residual is calculated as the difference between pCO$_2$ and the calculated pCO$_2$-T:
$$\text{pCO}_2\text{-Residual} = \text{pCO}_2 - \text{pCO}_2\text{-T}$$
Prior to algorithm processing, pCO$_2$-Residual values > 250 µatm and < -250 µatm from the
testbed were filtered out to target values that are not representative of the real ocean. These pCO$_2$-
Residual values generally correspond to high pCO$_2$, above the maximum value in SOCAT (816
µatm; Stamell et al., 2020). The excluded data points (less than 0.2 % per member) mostly occurred



in output from the CanESM2 model, and were restricted geographically, predominantly along the
western coastline of South America.

171         The eXtreme Gradient Boosting method (XGB; Chen and Guestrin, 2016) is used to

develop an algorithm that allows driver variables (i.e., SST, SSS, Chl-a, MLD, $xCO_2$) to predict
the $pCO_2$-Residual (**Fig. 1**; Step 2). The $pCO_2$-Residual and associated feature variables is split
into validation, training and testing sets. The test and validation set each account for 20 % of the
data, leaving 60 % for training. The validation set is used to optimize the algorithm
hyperparameters, which define the architecture of decision trees used in the model. The training
set is used to build the decision trees in XGB, while the test set is used to evaluate the performance
of the final algorithm. The XGB algorithm for this study used 4,000 decision trees with a maximum
depth of 6 levels. For the final reconstruction of surface ocean $pCO_2$ across all space and time
points, the previously calculated $pCO_2$-T values are added back to the reconstructed $pCO_2$-
Residual (**Fig. 1**; Step 3).

182         The full XGB process, including 1) training/evaluating/testing and 2) reconstructing

globally at a monthly resolution, was repeated individually for each LET member. This process
provided therefore a total of 75 unique reconstruction vs. 'model truth' pairs, which can be
statistically compared (**Fig. 1**; Step 4).
*2.3 Statistical Analysis in the Testbed*
The statistical comparisons between the test set and the reconstructions are equivalent to what
would be derived using real-world data ('seen' values). Since we are using a testbed, we can also
include comparisons on additional independent data, referred to as 'unseen' values, which
represent the 1°x1° grid cells of the ensemble members that do not correspond to SOCAT or
Saildrone USV observations. A suite of statistical metrics can be used to compare the
reconstruction to the 'model truth' in order to assess how well the algorithm can extrapolate from
sparse data to full-field coverage (**Fig. 1**; Step 4). In this study, we focus on bias and root-mean-
squared error (RMSE). Bias is calculated as 'mean prediction – mean observation' (i.e., $pCO_2$
predicted by XGB subtracted by the $pCO_2$ 'model truth'), and is a measure of over- or
underestimation in the reconstructions. RMSE measures the magnitude of the predicted error and
is calculated as the square root of the mean of the squared errors.





*2.4 Overview of sampling patterns and model runs*
First, we sampled target and driver variables from the LET based on sampling distributions
equivalent to that of the SOCAT database ('SOCAT baseline'). Then, we combined the 'SOCAT
baseline' with testbed output representing additional Saildrone USV coverage in the Southern
Ocean. The additional Southern Ocean coverage was based on 1) the Sutton et al. (2021) sampling
campaign from 2019 ('one-latitude' track) and 2) potential future meridional USV observations
('zigzag' track) (**Fig. 2**). We performed a total of 10 experimental runs (**Table 1**). These represent
different sampling approaches, including: 1) repeating USV sampling over a five- or ten-year
period, 2) varying the number of USVs and thus the total number of observations, and 3) restricting
all observations to southern hemisphere winter months. By comparing the different runs, we can
assess whether or not certain targeted sampling strategies in the Southern Ocean can improve
surface ocean $pCO_2$ ML reconstructions. As discussed above, the LET runs to 2016 only (Gloege
et al., 2021). Saildrone USV observations were therefore sampled from the testbed starting in year
2006 or 2007 (for the ten-year sampling) or 2012 (for the five-year sampling) until 2016, i.e., the
final year of the testbed.
*2.4.1 'One-latitude' runs*
Six out of the ten experimental runs include the 'one-latitude' track (**Table 1**). The 2019 Saildrone
USV journey (Sutton et al., 2021) covered an 8-month period, from January to August. Since the
USV was recovered in early August, it did not cover the entire southern hemisphere winter (**Fig.**
**S1**). We repeated this 'one-latitude' eight-month sampling pattern for five years ('5Y_J-A'; 2,075
observations) and ten years ('10Y_J-A'; 4,150 observations). In order to evaluate year-round
('YR') coverage, the eight-month sampling period (January-August) was shifted by one month
each year for ten years ('10Y_YR'; 4,150 observations). Furthermore, in order to evaluate the
impact of increased sampling, the 2019 Saildrone USV track was repeated 12 times with
incremental offsets of 1° from the original track, covering an additional 6° north and south (**Fig.**
**S2**). This 'high-sampling'-run ('x13_10Y_J-A'; 44,250 observations) represents a total of 13
USVs. We also performed an additional 13 USV run, but including observations from southern
hemisphere winter ('W') months only ('x13_10Y_W'; 25,395 observations). Finally, considering
the cost of deploying 13 USVs, a downscaled 'multiple-USV-winter-only'-run was tested,



including five USVs sampling over a period of five years ('x5_5Y_W'; 5,022 observations). This
run covers an additional 2° north and south from the original USV track.
*2.4.2 'Zigzag' runs*
Four of the ten experimental runs represent potential meridional sampling in the Southern Ocean
('zigzag' tracks; **Table 1**) as suggested by Djeutchouang et al. (2022). Due to limited solar
radiation that powers the Saildrone USVs, we let the sampling occur at a maximum latitude of 55°
S. This alternative sampling pattern represents USVs sailing west to east in a north/south 'zigzag'
pattern covering 40° S and 55° S for every 30° of longitude (**Fig. 2**). We created two scenarios.
For the first scenario, every 30° of longitude from 40° S and 55° S is visited every three months
within a single year as suggested by Lenton et al. (2006). Considering the average Saildrone USV
speed, this scenario represents four platforms equally spaced around the Southern Ocean. This
sampling pattern was repeated for 10 years, with year-round coverage ('Zx4_10Y_YR'; 7,600
observations), and for southern hemisphere winter months only ('Zx4_10Y_W'; 2,500
observations). The second scenario represents a 'high-sampling' strategy, where every 30° of
longitude from 40° S and 55° S is visited approximately monthly. This can be achieved by
deploying 10 platforms equally spaced around the Southern Ocean. This sampling pattern is
repeated for five years, sampling year-round ('Z_x10_5Y_YR'; 11,400 observations) and during
southern hemisphere winter months only ('Z_x10_5Y_W'; 3,800 observations).

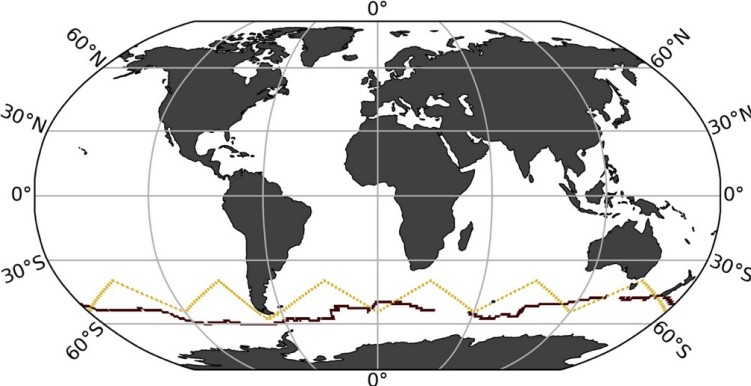


**Figure 2:** Saildrone Uncrewed Surface Vehicle (USV) tracks representing the first circumnavigation around
Antarctica from 2019 in maroon ('one-latitude' track; Sutton et al., 2021) and an alternative virtual route with
meridional coverage ('zigzag' track).



| Run name | 5Y_J-A | 10Y_J-A | 10Y_YR | x13_10Y_J-A | x13_10Y_W | x5_5Y_W | Z_x4_10Y_YR | Z_x4_10Y_W | Z_x10_5Y_YR | Z_x10_5Y_W |
|---|---|---|---|---|---|---|---|---|---|---|
| Saildrone track | One-lat | One-lat | One-lat | One-lat | One-lat | One-lat | Zigzag | Zigzag | Zigzag | Zigzag |
| Years of sampling | 5 | 10 | 10 | 10 | 10 | 5 | 10 | 10 | 5 | 5 |
| # of Saildrones | 1 | 1 | 1 | 13 | 13 | 5 | 4 | 4 | 10 | 10 |
| Duration of sampling | Jan-Aug | Jan-Aug | Year-round | Jan-Aug | SO winter | SO winter | Year-round | SO winter | Year-round | SO winter |
| Total observations | 2,075 | 4,150 | 4,150 | 44,250 | 25,395 | 5,022 | 7,600 | 2,500 | 11,400 | 3,800 |
| Global coverage increase (%) | 0.01 | 0.02 | 0.02 | 0.2 | 0.1 | 0.02 | 0.03 | 0.01 | 0.04 | 0.01 |

**Table 1.** Overview of the different Saildrone USV sampling patterns tested in this study using the XGBoost Machine Learning algorithm (Gloege et al., 2021; Bennington et al., 2022a) to estimate surface ocean $pCO_2$. The 'one-latitude' ('one-lat') track incorporate the Saildrone USV route from Sutton et al. (2021), while the 'zigzag' track represents potential future meridional sampling (see **Fig. 2**). The total number of USV observations (in bold) represent 1°x1° monthly Saildrone USV observations. J-A= January-August. YR = year-round. W = southern hemisphere winter. x4, x5, x10 and x13 = four, five, ten and 13 USVs. SO winter = Southern Ocean winter months, i.e., June, July, August and also including September. Note that all runs also included SOCAT coverage.

*2.5 Air-sea $CO_2$ flux*

To assess the global ocean carbon sink associated with our $pCO_2$ reconstructions, air-sea $CO_2$ exchange was calculated. Here, we computed air-sea $CO_2$ fluxes using the bulk formulation with python package Seaflux.1.3.1 (https://github.com/lukegre/SeaFlux; Gregor et al. 2021; Fay et al., 2021). We calculated global and Southern Ocean flux in the same manner for 1) the testbed 'model truth', 2) the SOCAT baseline and 3) the 10 experimental USV runs.

The net sea–air $CO_2$ flux was estimated using:

$$Flux = k_w \cdot sol \cdot (pCO_2^{ocn} - pCO_2^{atm}) \cdot (1 - ice)$$

where '$k_w$' is the gas transfer velocity, 'sol' is the solubility of $CO_2$ in seawater (in units of mol m$^{-3}$ µatm$^{-1}$), '$pCO_2^{ocn}$' is the partial pressure of surface ocean carbon (in µatm), either from the 'model truth' or from the reconstructions, and $pCO_2^{atm}$ (in µatm) is the partial pressure of atmospheric $CO_2$ in the marine boundary layer. For GFDL, we used direct model output of $pCO_2^{atm}$, while for CESM and CanESM2, $pCO_2^{atm}$ was calculated individually, as the product of surface $xCO_2$ and sea level pressure ($pCO_2^{atm}$ from CESM was corrected for the contribution of water vapor pressure). Finally, to account for the seasonal ice cover in high latitudes, the fluxes were weighted by 1 minus the ice fraction ('ice'), i.e., the open ocean fraction. Inputs to the calculation include EN4.2.2 salinity (Good et al., 2013), SST and ice fraction from NOAA Optimum Interpolation Sea Surface Temperature V2 (OISSTv2) (Reynolds et al., 2002), and





surface winds and associated wind scaling factor from the European Centre for Medium-Range
Weather Forecasts (ECMWF ERA5 sea level pressure (Hersbach et al., 2020). Results presented
show the global and Southern Ocean (< 35° S) fluxes in units of Pg C yr$^{-1}$.
Note that, reconstructions of $pCO_2$ for the SOCAT baseline and the experimental USV runs
are limited in their spatial extent to the open ocean (see **Sect. 2.1**; excluding coastal areas, the
Arctic Ocean and marginal seas). The same mask was thus also applied when calculating the flux
of the 'model truth', prior to comparison with the reconstructions.
**3. Results**
*3.1 Performance metrics for the 'SOCAT baseline' reconstruction*
The mean bias for the entire testbed period (i.e., 1982-2016) is 0.63 µatm globally (**Fig. 3a**) and
1.4 µatm for the Southern Ocean (< 35° S; **Table S1**). Bias is much closer to zero for mid- (between
35° S and 35° N; 0.23 µatm) and northern latitudes (> 35° N; 0.11 µatm) (**Fig. 3a**). There is a
significant difference in bias considering southern hemisphere winter months (June, July, August)
versus summer months (December, January, February), with a global mean bias (for 1982-2016)
of 1.3 µatm compared to 0.07 µatm, respectively (**Table S1**), due to the sparseness of SOCAT
observations from the southern hemisphere during the harsh winter season (**Fig. S3a**). The mean
RMSE for the entire testbed period (i.e., 1982-2016) is 7.9 µatm globally (**Fig. 3b**) and 8.5 µatm
for the Southern Ocean (**Table S1**). RMSE is highest in the Eastern Tropical and Southeastern
Pacific Ocean and in the Southern Ocean, where the algorithm generally overestimates $pCO_2$ (i.e.,
positive bias; **Fig. 3a**). This is consistent with the areas significantly undersampled by SOCAT
(**Fig. S3b**). Except for these areas, RMSE and bias is generally low (close to zero) in the open
ocean, but show higher values along coastlines (**Fig. 3b**).

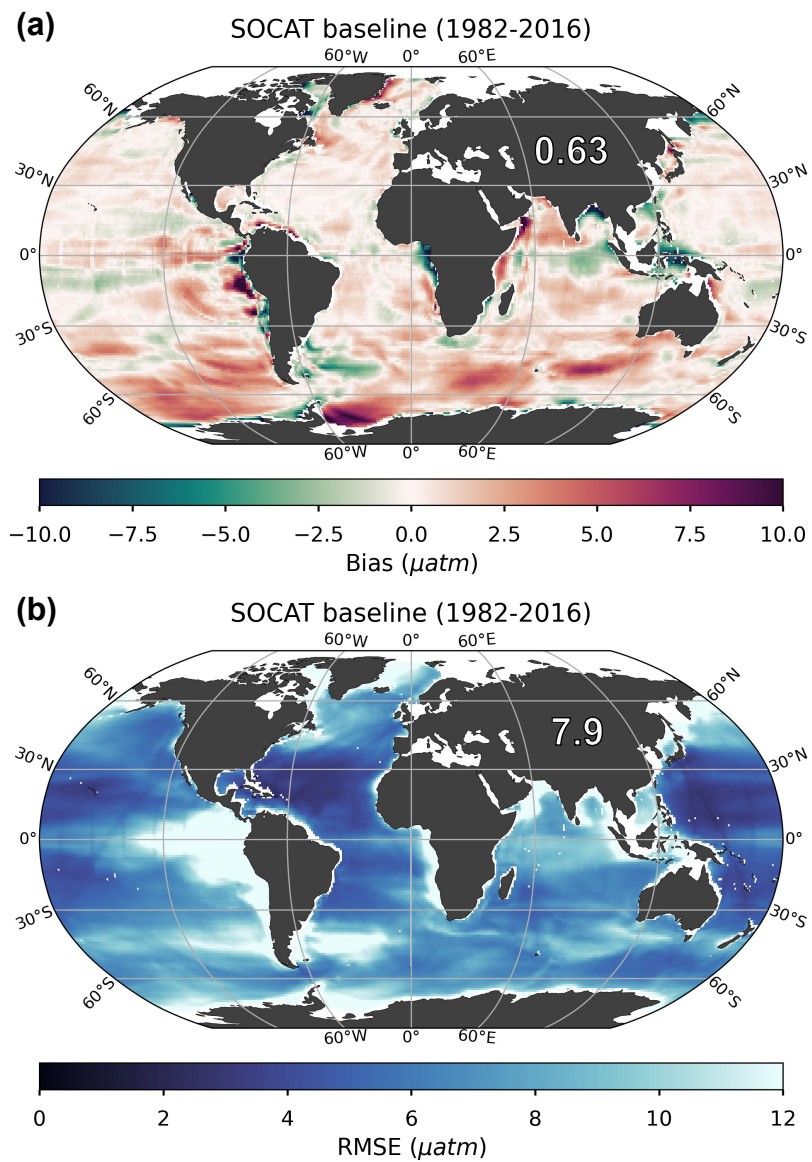

**Figure 3:** Bias (**a**) and root-mean-squared error (RMSE) (**b**) when comparing the baseline machine learning
reconstruction with the testbed 'model truth', averaged over the 75 ensemble members for the period of 1982 through
2016. The testbed was sampled based on SOCAT observations only (i.e., no USV). The global mean bias and RMSE
is 0.63 µatm and 7.9 µatm, respectively. Red and green areas in **a** indicate regions where the reconstruction is biased
high (i.e., overestimates $pCO_2$) and low (i.e., underestimates $pCO_2$), respectively. Generally, RMSE is highest in the
East and South Pacific Ocean and in the Southern Ocean, where the algorithm also generally overestimates $pCO_2$
(positive bias; **a**). Note that only the open ocean was considered in the reconstruction, so several areas were masked
out prior to algorithm processing, such as the Arctic Ocean, coastal areas and marginal seas (no data; white areas in
figures).






### 3.2 Reconstruction improvements with Saildrone USV additions

Our presentation of global maps is limited to runs 'x5_5Y_W' (5,022 observations) and 'Z_x4_10Y_YR' (7,600 observations). These runs were selected as they represent observational schemes that are realistic in the near-term future considering logistics and cost level, both non-meridional and meridional sampling, and different approaches to observing duration and seasonal coverage. For the remaining runs, equivalent maps can be found in the **Supplement**.

### 3.2.1 Bias

All Saildrone USV runs show a reduction in bias compared to the global mean 1982-2016 SOCAT baseline (**Figs. 4a**, **S4**). The improvement in bias is mainly due to lower reconstructed $pCO_2$ values at southern latitudes, where the baseline reconstruction generally overestimates $pCO_2$ (**Fig. 3a**). The global mean bias for 'zigzag' run 'Z_x4_10Y_YR' is 0.51 µatm, a higher improvement (19 %) over the SOCAT baseline compared to the 'one-latitude' run 'x5_5Y_W' (11 % improvement; mean bias = 0.57 µatm;) (**Fig.4a**; **Table S1**). Generally, the 'zigzag' runs show higher improvements from the SOCAT baseline (19-31 % improvement; mean bias = 0.44-0.51 µatm) compared to the 'one-latitude' runs (7-19 % improvement; mean bias = 0.52-0.59 µatm) (**Fig S4**; **Table S1**). However, the 'one-latitude'-run 'x13_10Y_W' that samples southern hemisphere winter months only, stands out with the lowest global mean bias of 0.39 µatm, representing a 39 % improvement from the SOCAT baseline (**Table S1**; **Fig. S4**). This run, however, has three or five times more observations (25,395) than 'Z_x4_10Y_YR' and 'x5_5Y_W', respectively.

Compared to the entire testbed period, even larger improvements in global mean bias are shown for the period of Saildrone USV additions (2006-2016 and 2012-2016; **Figs. 4a** vs. **4b**, **Figs. S4** vs. **S5**). Compared to the SOCAT baseline, run 'x13_10Y_W' results in a bias improvement of 95 %, while the remaining 'one-latitude' runs and the 'zigzag' runs show improvements up to 63 % and 85 %, respectively (**Fig. S5**).

Perhaps surprisingly, there is not a strong connection between the global or Southern Ocean mean bias and the number of added USV observations (**Fig. 5**). The 'one-latitude' 'high-sampling' run 'x13_10Y_J-A' (44,250 observations) show similar bias or is outperformed by all 'zigzag' runs as well as the 'one-latitude'-runs that restrict sampling to southern hemisphere winter months (i.e., 'x5_5Y_W' and 'x13_10Y_W').




Considering the change in bias from year-to-year, the SOCAT baseline shows positive bias
at all latitudes in the beginning of the testbed period, before improvement occurs around year 1990
(**Fig. 6a**). This is consistent with increasing SOCAT sampling with time for the time period
considered here (i.e., up to 2016; **Fig. S3c**). As SOCAT observations are biased towards the
northern hemisphere (**Fig. S3a, b**), bias in the Southern Ocean (< 35° S) increases significantly
starting in 2000s and remains high until the end of the testbed period (**Fig. 6a**). By adding USV
sampling, bias in the Southern Ocean improves over the SOCAT baseline around year 2000 (**Fig.**
**6b-d**; **Fig. S6**), up to 6-12 years prior to the introduction of additional samples in either 2006 or
2012. Run 'Z_x10_5Y_W', which has the lowest bias out of the 'zigzag' runs (**Fig. 5**), shows
improvement even further back in time, until the beginning of the testbed period (**Fig. S6**). While
the annual mean bias of the 'zigzag' runs vary in a similar manner, there is a large spread between
the 'one-latitude' runs (**Fig. 6d**).



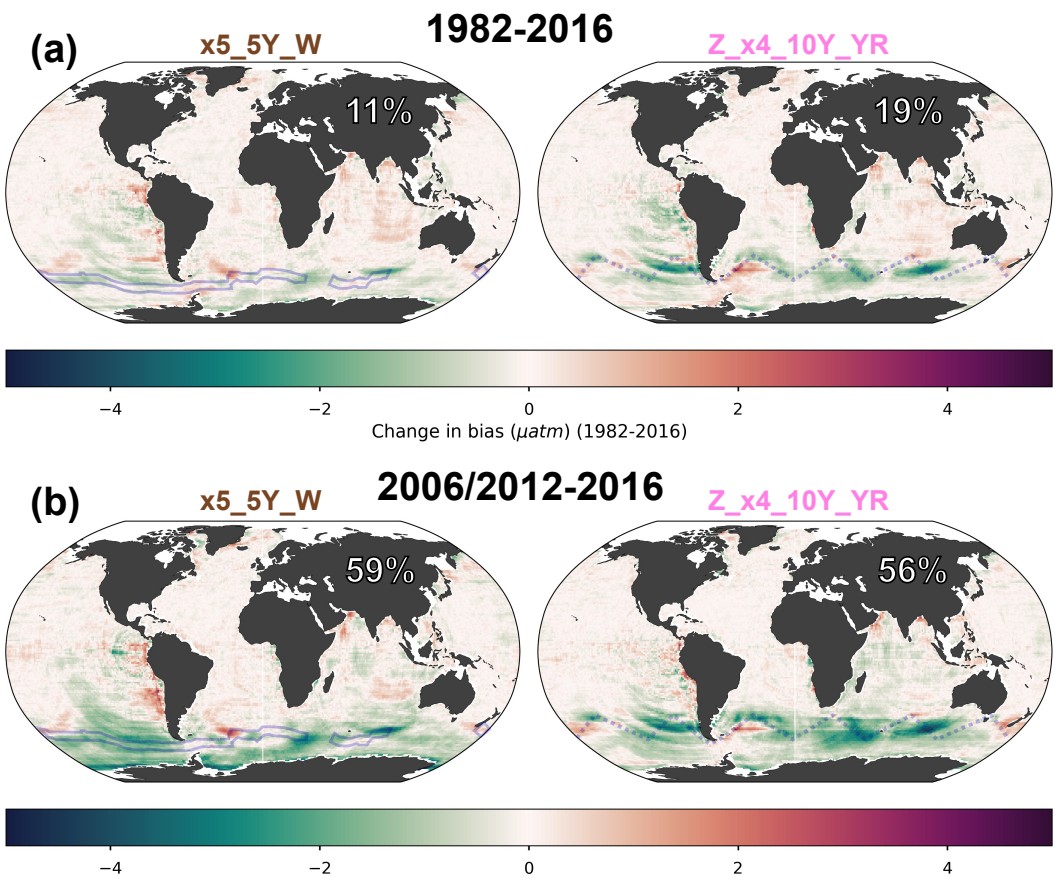

Figure 4: Change in bias when comparing run 'x5_5Y_W' and 'Z_x4_10Y_YR' to the SOCAT baseline reconstruction, averaged over the duration of the testbed period (**a**; 1982-2016) and the period of USV additions (**b**; 2006-2012 or 2012-2016). Negative change in bias is found across the southern latitudes, indicating an improvement compared to the SOCAT baseline that overestimates $pCO_2$ (**Figure 3a**). The percent global improvement is shown on each panel. Note that improvement is greater in the period of Saildrone USV additions compared to the entire testbed period.



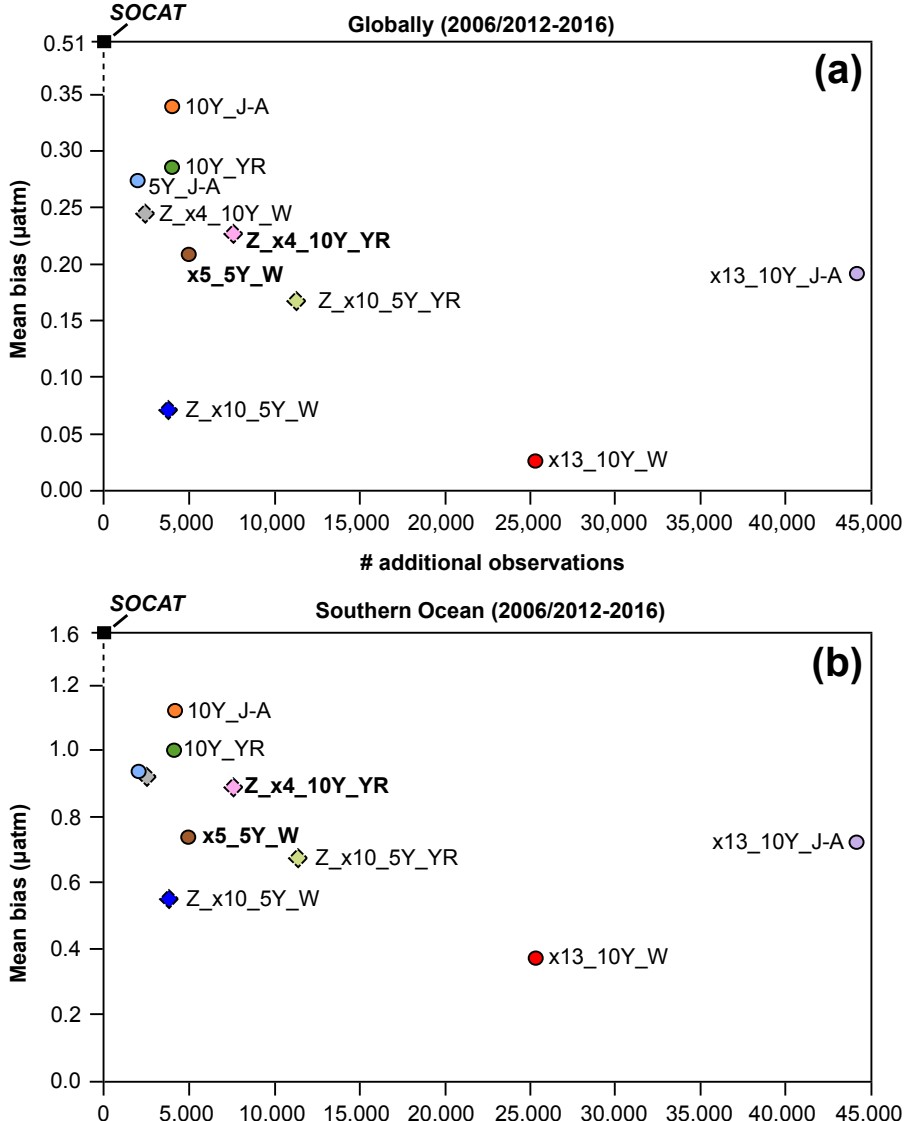

**Figure 5:** Mean bias globally (**a**) and for the Southern Ocean (**b**) for the duration of Saildrone USV sampling (2006-2016 or 2012-2016) for all runs presented in **Table 1**. Circles represent runs using the 'one-latitude' track (Sutton et al., 2021), while diamonds represent 'zigzag' runs. Runs highlighted in bold correspond to the two selected runs mapped in **Figure 4, 6, 7** and **9**. Global (0.51 µatm) and Southern Ocean (1.6 µatm) bias values shown for the SOCAT baseline (black squares) represent a mean of values for 2006-2016 (global = 0.52 µatm, S. Ocean = 1.63 µatm) and 2012-2016 (global = 0.51 µatm, S. Ocean = 1.56 µatm). The SOCAT baseline run included 261,733 monthly 1°x1° observations. Overall, there is not a strong correlation between bias and the number of observations, or duration of sampling.







**Figure 6:** Zonal mean, annual mean Hovmöller of bias in SOCAT baseline with the testbed 'model truth', average of 75 ensemble members (**a**). There is a positive bias at all latitudes from 1982-1995; bias drops to around zero in the late 1980s; and then, particularly in the Southern Ocean, increases at 2000 and remains high through 2016. Change in bias of run 'x5_5Y_W' (**b**) and 'Z_x4_10Y_YR' (**c**)compared to the SOCAT baseline reconstruction shown in (**a**). Negative changes in the Southern Ocean represents an improvement. The improvement in bias expands back in time well beyond the duration of USV additions for both runs (shown by arrows on each panel). Annual mean bias for the Southern Ocean (> 35° S) for all runs(**d**). There is a large spread in the impact on bias with 'one-latitude' USV sampling (solid lines), while the 'zigzag' runs (dashed lines) more consistently reduce bias.

*3.2.2 Root-mean squared error (RMSE)*

Similar to bias, improvements in RMSE are most significant during the period of USV additions and within the Southern Ocean (**Fig. 7a** vs. **7b**). For the duration of USV additions, the 'one-latitude' runs show improvements in global mean RMSE of 1-4 % (0.3-2 % for 1982-2016), while the 'zigzag' runs show higher improvements between 3-8 % (2-3 % for 1982-2016) (**Figs. 7**, **S7**, **S8**). RMSE is further reduced in southern hemisphere winter in the Southern Ocean by up to 26 % (mean RMSE of 6.9 μatm; **Table S1**). There is minimal change in RMSE (or bias) during southern hemisphere summer months (DJF; **Fig. S9**). The two 'zigzag' runs sampling year-round ('Z_x4_10Y_YR' and ''Z_x10_5Y_YR) have the lowest RMSE values both globally and in the Southern Ocean (**Fig. 8**).

The 'zigzag' runs, as well as the 'high-sampling' 'one-latitude'-runs (i.e., 'x13_10Y_J-A' and 'x13_10Y_W'), show improvements compared to the SOCAT baseline from the initiation of sampling (**Figs. 9**, **S10**). The year-round 'zigzag' runs, however, show improvement in the Southern Ocean from the beginning of the testbed period (**Figs. 9c**, **d**, **S10**). RMSE improvements back in time are more significant for all runs in the southern hemisphere winter months (**Fig. S11**).



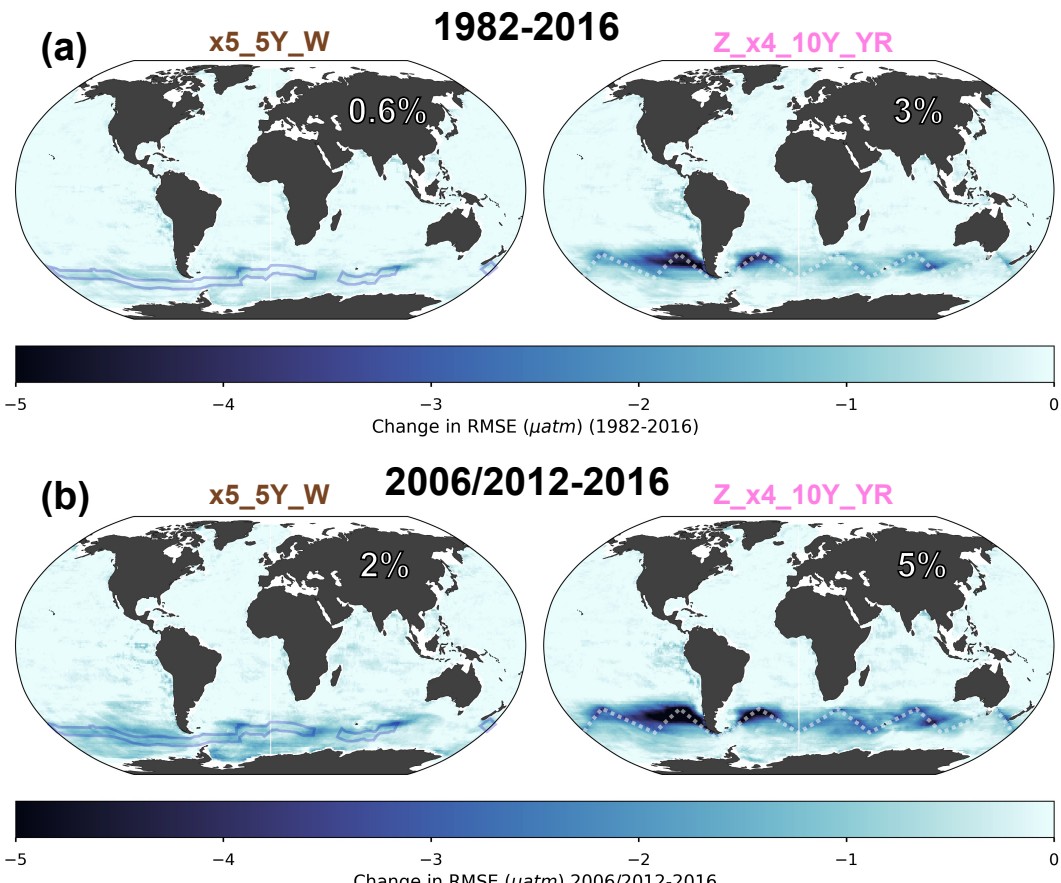

**Figure 7:** Change in RMSE when comparing run 'x5_5Y_W' and 'Z_x4_10Y_YR' to the SOCAT baseline reconstruction, averaged over the duration of the testbed period (**a**; 1982-2016) and the period of Saildrone USV additions (**b**; 2006-2012 or 2012-2016). Improvement in RMSE occurs mainly in southern latitudes (<35°S), where the baseline reconstruction shows high RMSEs (**Fig. 3b**). The percent global improvement is shown on each panel. Note the greater improvement for the period of USV additions compared to the entire testbed period.



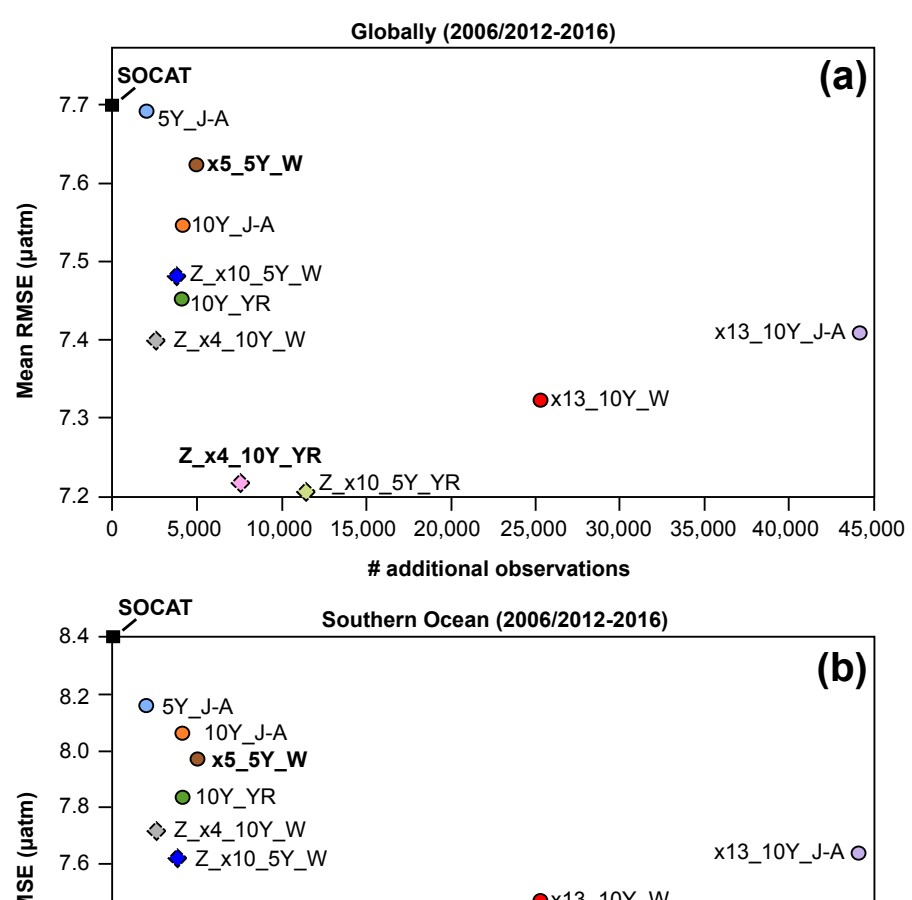

**Fig. 8:** Mean RMSE globally (**a**) and for the Southern Ocean (< 35° S; **b**) for the duration of Saildrone USV sampling (2006-2016 or 2012-2016) for all runs presented in **Table 1**. 'One-latitude' runs (circles), 'zigzag' runs (diamonds). Runs highlighted in bold correspond to the two selected runs mapped in **Figure 4, 6, 7** and **9**. Global (7.7 μatm) and Southern Ocean (8.4 μatm) bias values shown for the SOCAT baseline (black squares) represent a mean of values for 2006-2016 (global = 7.6 μatm, S. Ocean = 8.3 μatm) and 2012-2016 (global = 7.8 μatm, S. Ocean = 8.5 μatm). The SOCAT baseline run included 261,733 monthly 1°x1° observations. Overall, there is not a strong correlation between increasing number of observations or duration of sampling and decreasing RMSE.



407





**Figure 9:** Zonal mean, annual mean Hovmöller of RMSE in SOCAT baseline with the testbed 'model truth', average
of 75 ensemble members (**a**). Dark and light areas represent regions where RMSE is low and high, respectively. RMSE
is highest at latitudes > 60° S, > 60° N and around 40° S and the equator. RMSE is higher at all latitudes in the
beginning of the testbed period, before some improvement occurs in the 1990s. Change in RMSE of run 'x5_5Y_W'
(**b**) and 'Z_x4_10Y_YR'(**c**) compared to the SOCAT baseline reconstruction shown in (**a**). Dark areas represent
regions where the change in RMSE is negative, i.e., where the Saildrone USV sampling additions improve the pCO₂
reconstruction. Run 'Z_x4_10Y_YR' shows improvements in RMSE within the Southern Ocean, which expand well
beyond the duration of Saildrone USV additions (shown by arrow on panel). Annual mean RMSE for the Southern
Ocean (> 35° S) for all runs (**d**).

*3.3 Impact on the air-sea CO$_2$ flux with Saildrone USV additions*

Air-sea flux was calculated in the same manner for both the ML reconstructions and the 'model truth', which allows for direct comparison of the differences in fluxes (see **Sect. 2.5**). These flux estimates are made to inform understanding of the errors that may exist in CO$_2$ flux estimates derived from pCO$_2$ reconstructions, and how new sampling could address these errors. These fluxes are not estimates of real-world fluxes.

Compared to the 'model truth', the SOCAT baseline reconstruction underestimates the global and Southern Ocean sink by 0.11-0.13 Pg C yr$^{-1}$ over 1982-2016 (**Fig. 10**; **Table S2**). Regardless of sampling pattern, adding Saildrone USV observations increases both the global and Southern Ocean mean sink compared to the SOCAT baseline (**Figs. 10**, **S12**). The 'one-latitude' runs show an increase of 0.01-0.03 Pg C yr$^{-1}$ (2-6 % strengthening) of the Southern Ocean sink (1982-2016), while the 'zigzag' runs lead to an even stronger sink 0.04-0.06 Pg C yr$^{-1}$ (7-11 % strengthening) (**Table S3**). When averaging over the years of Saildrone USV sampling addition (i.e., 2006-2012 and 2012-2016), the Southern Ocean sink increases up to 0.09 Pg C yr$^{-1}$ (14 % strengthening) for the 'one-latitude' runs and up to 0.1 Pg C yr$^{-1}$ (15 % strengthening) for the 'zigzag' runs (**Table S3**). These same features are found for the global ocean (**Fig. S12**; **Table S3**).

All of the 'zigzag' runs quite closely match both the global and Southern Ocean 'model truth' air-sea CO$_2$ flux for the duration of sample additions (**Figs. 10**, **S12**). Except for the first couple of years of sample addition for the 'high-sampling'-run 'x13_10Y_J-A', none of the 'one-latitude' runs are able to match the 'model truth' air-sea CO$_2$ flux, as they all underestimate the flux (**Figs. 10, S12**). The 'zigzag' runs have impact on the air-sea flux from an earlier date, starting



to pull the results away from the SOCAT baseline and toward the 'model truth' already in the late-
1990s, while the 'one-latitude' runs do the same about a decade later (**Figs. 10**, **S12**).

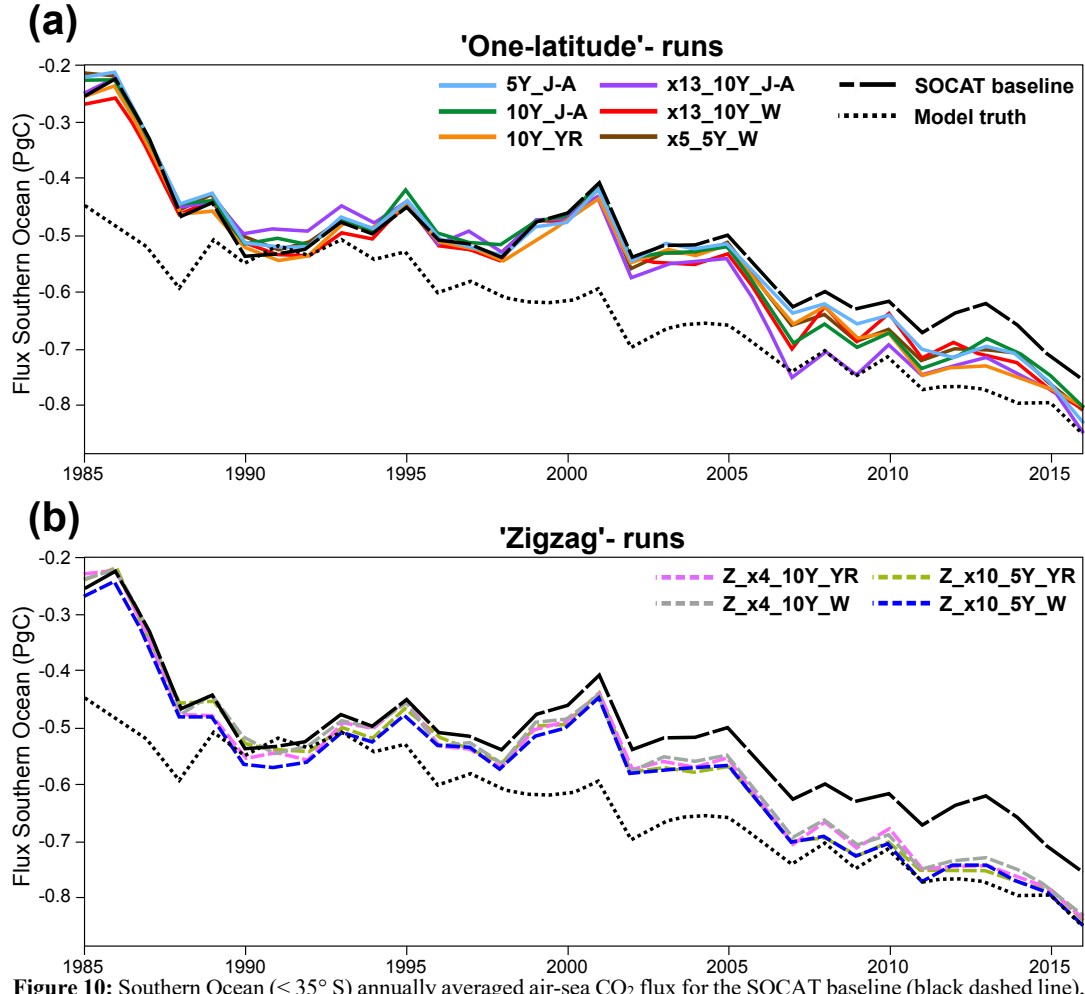

**Figure 10:** Southern Ocean (< 35° S) annually averaged air-sea CO$_2$ flux for the SOCAT baseline (black dashed line),
'model truth' (black dotted line) 'one-latitude' runs (**a**; solid lines) and 'zigzag' runs (**b**; dashed lines), averaged over
the 75 ensemble members. Compared to the SOCAT baseline, regardless of sampling pattern, the Saildrone USV
additions lead to an increased ocean sink. The 'zigzag' runs generate a stronger sink compared to the 'one-latiude'
runs, and closely match the 'model truth' for the duration of sample additions.


**4. Discussion**
We have tested the pCO$_2$-Residual reconstruction method with the Large Ensemble Testbed (LET)
to estimate its fidelity and understand how new samples could increase skill. We find that,



regardless of the chosen Saildrone USV sampling pattern, the reduction in both bias and RMSE
compared to the SOCAT baseline is most prominent within the Southern Ocean (< 35° S) during
the period of which Saildrone USV observations were added (**Figs. 4, 6, 7, 9**). However, it is
important to mention that additional Southern Ocean sampling also improves $pCO_2$ reconstructions
globally (**Figs. 5a**, **8a**). Based on our experiments, a combination of factors seems to be important
in order to improve both the global and Southern Ocean $pCO_2$ reconstructions, and include mainly
the type of sampling pattern and seasonality of sampling, but also to some extent the number of
additional observations. Importantly, increasing the number of observations or duration of
sampling (5 vs. 10 years) is not the sole determining factor for improving the reconstructions (**Figs.**
**5**, **8**). This is best demonstrated by the 'high-sampling'-run 'x13_10Y_J-A' (44,250 observations),
which does not provide significantly better reconstructions, or is even outperformed, by runs with
2-18 times less observations, but that cover the full southern hemisphere winter (**Figs. 5**, **6d**, **8**,
**9d**). Run 'x13_10Y_J-A' does not include more than a few observations in the month of August,
as it follows the temporal pattern of the real-world 'one-latitude' Saildrone USV expedition (**Fig.**
**S1**; Sutton et al., 2021). The 'one-latitude' runs '10Y_J-A' and '10Y_YR' are directly comparable
in terms of sample duration, spatial extent and number of observations (**Table 1**), but the latter,
which covers all months, always shows lower RMSE and bias (**Figs. 5**, **6d**, **8**, **9d**). These examples
attest to the importance of addressing the issue of significant undersampling in the Southern Ocean
during the winter season (**Figs. S3a, b**).

472        Another important comparison is the 'one-latitude'-run 'x5_5Y_W' (5,022 observations)

and 'zigzag'-run 'Z_x10_5Y_W' (3,800 observations) that both sample during southern
hemisphere winter months over a five-year period (**Table 1**), where the 'zigzag'-run consistently
performs better even though it includes fewer observations (**Figs. 5**, **8**). Most of the runs that
perform similar to, or outperform, the above-mentioned 'high-sampling'-run 'x13_10Y_J-A'
(44,250 observations), sample in a 'zigzag' pattern. Out of all 10 runs, the 'year-round' 'zigzag'
runs ('Z_x4_10Y_YR' and 'Z_x10_5Y_YR') are most able to reduce the magnitude of error as
shown by the lowest RMSE values (**Figs. 8**, **9d**). A recent study performed similar sampling
experiments as shown here, by comparing sampling from different types of autonomous platforms
to a SOCAT baseline (Djeutchouang et al., 2022). They emphasized the importance of capturing
the significant differences in $pCO_2$ that exist across meridional gradients during summer and
winter months (up to 15 µatm; Djeutchouang et al., 2022). The meridional coverage provided by





the 'zigzag' runs could explain why these runs generally outperform the 'one-latitude' runs in our
study, and show significant reduction in both RMSE and bias, even though the global $pCO_2$ data
density is raised by as little as 0.01-0.04 %.
The greatest reduction in bias out of all runs is however shown by run 'x13_10Y_W' (**Figs.**
**5**, **6d**), which represents 'one-latitude' 'high-sampling' (i.e., 25,395 observations) during southern
hemisphere winter months only. This sampling strategy seems thus to have a higher ability to
reduce the ML model's tendency to overestimate $pCO_2$ in the Southern Ocean compared to any of
the meridional ('zigzag') runs. However, it should be noted that run 'x13_10Y_W' cover areas
south of 55° S (**Fig. S2**), and its improvement in bias (and RMSE) is particularly prevalent at such
high latitudes (e.g., **Figs. S5**, **S6**, **S8**, **S10**). Whether or not this run is in fact feasible with current
or future technology is uncertain as parts of the southernmost tracks cover the Southern Ocean ice
zone (**Fig. S13**), and solar radiation for solar-powered platforms and sensors becomes very limited
during winter south of 55° S. Furthermore, this particular sampling strategy requires 13 USVs, and
thus would be the most costly of the observing scenarios. Although run 'x13_10Y_W'
demonstrates the highest reduction in bias out of all runs, the 'zigzag' runs still reduce bias in the
Southern Ocean by 44-65 % (vs. 77 % for run 'x13_10Y_W').
Overall, the 'zigzag' runs include significantly fewer observations, require less USVs,
collect samples over the same duration, or even half the time as run 'x13_10Y_W', cover areas
north of 55°S and within the ice-free zone, and show major improvement in the reconstruction of
$pCO_2$, attested to by reductions in both bias and RMSE. The 'zigzag' runs also closely match both
the global and Southern Ocean 'model truth' air-sea $CO_2$ flux for the duration of sample additions
(**Figs. 10**, **S12**). It also appears that the 'zigzag' runs generally have a greater impact on both the
$pCO_2$ reconstruction and the air-sea flux further back in time, starting to deviate from the SOCAT
baseline earlier compared to the 'one-latitude' runs (**Figs. 6**, **9**, **10**, **S6**, **S10**, **S11**, **S12**). Even the
'zigzag' scenarios with the least number of USVs (e.g., 'Z_x4_10Y_YR') reduces Southern Ocean
reconstruction bias and RMSE by up to 46 % and 13 %, respectively, and could provide a basis
for realistic future Southern Ocean $pCO_2$ sampling campaigns.
The main motivation for improving surface ocean $pCO_2$ reconstructions is so that we can
more accurately estimate the current and future oceanic uptake of anthropogenic carbon. The
Southern Ocean is a significant carbon sink, but estimates of the air-sea $CO_2$ flux diverge





substantially in this region (Takahashi et al., 2009; Landschützer et al., 2014, 2015; Rödenbeck et al., 2015; Williams et al., 2017; Gray et al., 2018; Gruber et al., 2019; Bushinsky et al., 2019; Long et al., 2021; Fay and McKinley, 2021; Wu et al., 2022). Southern Ocean estimates incorporating observations from biogeochemical floats have shown a significantly weaker sink compared to those based only on observations from ships (Williams et al., 2017; Gray et al., 2018; Bushinsky et al., 2019). Bushinsky et al. (2019) performed similar sampling experiments as presented here, by comparing ML surface ocean $pCO_2$ reconstructions based on SOCAT alone vs. additional Southern Ocean floats. They showed that by adding the floats, the Southern Ocean carbon sink (mean of the period of float additions; 2015-2017) decreased (weakened) by 0.4 Pg C yr$^{-1}$. In contrast, by using a model testbed, we show that adding USVs increased (strengthened) the Southern Ocean and global ocean sink by up to 0.1 Pg C yr$^{-1}$ (**Figs. 10**, **S12**; **Table S3**), which is a significant fraction of the uncertainty in the global ocean carbon sink (0.4 Pg C yr$^{-1}$; Friedlingstein et al., 2022). Fed with real-world SOCAT data, the global mean air-sea flux estimate from the $pCO_2$-Residual method is similar to other available products (Bennington et al., 2022a), suggesting that other products may also underestimate the Southern Ocean carbon sink due to the spatio-temporal distribution of SOCAT data. Our experiments suggest that targeted USV observations could reduce this underestimation of the ocean carbon sink.

What else can we learn using the model testbed? The SOCAT baseline demonstrates a weakening of the global and Southern Ocean carbon sink in the 2000s (**Figs. 10**, **S12**), which is in agreement with various data products using real-world SOCAT data (e.g., Gruber et al., 2019; Landschützer et al., 2015; Bushinsky et al., 2019; Bennington et al., 2022; Gloege et al., 2022). Peaks in bias and RMSE coincide in time with the weakening sink (**Figs. 6d, 9d**). As shown by **Figure 10**, this 'low sink' is significantly exaggerated compared to the 'model truth'. To better understand this discrepancy, we performed an additional experiment based on run 'Z_x10_5Y_YR', but assumed sampling every year for the entire testbed period (i.e., 1982-2016). The results from this experiment show a significant reduction in the temporal variability of reconstruction bias; with the additional USV sampling, the reconstructed Southern Ocean air-sea $CO_2$ flux closely matches the 'model truth' for the entire testbed duration (**Fig. S14**). This suggests that the large decadal variability of air-sea $CO_2$ fluxes since the 1980s, and the weak anomaly in the Southern Ocean carbon sink in the early 2000s (Le Quéré et al., 2007; Landschützer et al., 2015; Gruber et al., 2019; Bennington et al., 2022a,b; Friedlingstein et al., 2022), may be at least





partially attributable to undersampling of the Southern Ocean. We will further explore this issue
in future work. Still, this preliminary experiment suggests that interpretations of trends and
variability of the global and Southern Ocean carbon sink should be considered with caution.
**5. Conclusions**
By using the Large Ensemble Testbed (LET), we show that targeted meridional and winter
sampling in the Southern Ocean can improve global and Southern Ocean ML surface ocean $pCO_2$
reconstructions. Significant improvements are possible by raising the global $pCO_2$ data density by
as little as 0.02-0.04 %. Further, we find that this modest amount of additional Saildrone USV
sampling increases the global and Southern Ocean air-sea $CO_2$ flux by up to 0.1 Pg C yr$^{-1}$, 25 %
of the uncertainty in the ocean carbon sink. Our findings are consistent with previous studies
suggesting that additional observations during southern hemisphere winter months and covering
meridional gradients can reduce uncertainties and biases in the reconstructions (Lenton et al., 2006;
Monteiro et al., 2010; Djeutchouang et al., 2022; Mackay et al., 2022). As opposed to other
autonomous platform approaches, Saildrone USVs obtain in situ $pCO_2$ observations with
uncertainties equivalent to the highest-quality observations collected by research ships (± 2 μatm;
Sabine et al., 2020; Sutton et al., 2021), and can operate at a high speed so that the spatial extent
and seasonal cycle of meridional gradients can be covered. The approach of combining high-
accuracy Saildrone USV and SOCAT observations represents thus a promising solution to improve
future surface ocean $pCO_2$ reconstructions and the accuracy of the ocean carbon sink. Lastly, we
show that the large variability in bias, and the weakening of the global and Southern Ocean carbon
sink in the 2000s, may be partially an artefact of Southern Ocean undersampling.
**Code availability**
Data analysis scripts will be made available in a GitHub repository upon publication.
**Data availability**
The Large Ensemble Testbed is publicly available at
https://figshare.com/collections/Large_ensemble_pCO2_testbed/4568555.

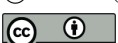

**Author contribution**

THH, GAM and AJS designed the experiments, and THH performed the simulations. THH, ARF and LG developed the code. THH and ARF calculated the air-sea fluxes. THH prepared the manuscript with contributions from all co-authors.

**Competing interests**

The authors declare that they have no conflict of interest.

**Acknowledgements**

We acknowledge funding from NOAA through the Climate Observations and Monitoring Program (Award #NA20OAR4310340) and from NSF through the LEAP STC (Award #2019625). This is PMEL contribution 5549. We would also like to acknowledge and thank Julius Busecke and Devan Samant for providing technical support.

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
