# Peer review of "Assessing improvements in global ocean pCO2 machine learning reconstructions with"

_Biogeosciences, 2023_

## Referee Comment (RC2)

**Comments on the manuscript entitled "Assessing improvements in global ocean $p\text{CO}_2$ machine learning reconstructions with Southern Ocean autonomous sampling"**

November 15, 2023

This study exploits Large Ensemble Testbed (LET) experiments targeting to prompt meridional and winter samples by Saildrone USVs in the Southern Ocean to improve the reconstruction of surface seawater partial pressure of $\text{CO}_2$ ($p\text{CO}_2$) and air-sea fluxes. For LET, 75 Earth System Models (ESM) have been selected to provide input ($p\text{CO}_2$ and potential driver variables) for a machine learning (ML)-based mapping approach. Two primary exercises have been conducted: ML-based reconstructions of $p\text{CO}_2$ with only SOCAT baseline and with Saildrone USVs sampling tracks added. Data reconstructions are evaluated with the model truth. In this manuscript, the authors have demonstrated that the reconstructions with additional USV data allow reducing the errors in $p\text{CO}_2$ and flux estimates. Despite appreciating the author's efforts in this study, Reviewer has not been convinced by its originality. Based on ESM output, numerous existing research works have shown additional data sampling (e.g., bgcArgo, SOCCOM, Sailboat,...) critical for error reduction in $p\text{CO}_2$ and flux estimation over the Southern Ocean and/or the global ocean [Bushinsky et al., 2019, Denvil-Sommer et al., 2021, Hauck et al., 2023, Landschützer et al., 2023]. One suggestion that would add value to the manuscript's findings is an analysis of spatial and temporal variations of flux estimates: to what extent their variability changes subject to the additional data. Some other major concerns are listed below.

1. Lines 149-153: "*To build reconstruction algorithms through the data-driven training that occurs in ML, the statistics in all other algorithms developed to date must identify a function that disentangles these competing effects of SST on pCO2. Here, the algorithm is assisted by removing this known temperature effect, and it must therefore only learn the pCO2 impacts from biogeochemical drivers*": there exist many other ML approaches [Friedlingstein et al., 2022] which do not separate the SST-effects from others on $p\text{CO}_2$ but succeeds in estimate $p\text{CO}_2$. The major concerns are how to assess the uncertainty derived from SST effect removal and impacts on the experiment outputs.

2. Figure 3: Relatively small bias and RMSE values have shown their imprints on the SOCAT track compared to "unseen" model truth. This evidences the problems of model overfitting. The authors can double-check whether model overfitting comes from the cross-validation technique or the $p$CO$_2$-Residual method. As the key findings of this manuscript are based on the data reconstruction results, Reviewer suggests the authors to carefully verify their methods and solve the problems of model overfitting before further consideration for publication.

**Editorial and specific comments**:

1. Lines 11-12: "anthropogenic" can be removed. The SO has taken up atmospheric CO$_2$ without specifying natural or anthropogenic sources.

2. Line 37: "$f$CO$_2$" is not defined. "uncertainty of $< 5$ $\mu$atm": this holds only for the measurements chosen to provide gridded SOCAT datasets.

3. Line 42: "Observation-based data products" $\longrightarrow$ "Data mapping methods".

4. Line 45: "These data products" $\longrightarrow$ "These methods".

5. Lines 46-47: please remove or change ";" in the brackets to facilitate reading. You can use "-" instead. Line 47: "$x$CO$_2$; atmospheric CO$_2$" $\longrightarrow$ "atmospheric CO$_2$ - $x$CO$_2$"

6. Line 48: "where these are co-located" $\longrightarrow$ "where their available data are co-located".

7. Lines 50-51: "*Since the data products rely on observations to train the algorithms and thus produce these relationships*": please rephrase this sentence. Data products do not train algorithms and produce relationships, but the ML-based methods themselves estimate the function between predictors and target data!

8. Line 57: "indirect $p$CO$_2$ estimates": can you define this term? Are they computed from float measurements of other carbonate variables?

9. Lines 67-68: "*Such improvements in sampling are critically important in the undersampled Southern Ocean*": USVs with low measurement uncertainty would prompt to be employed for observing network systems of $p$CO$_2$ but to draw this statement, it requires to provide the availability of USVs to sample $p$CO$_2$ by showing the sampling frequency and data coverage area over the SO?

10. Line 86: "actual observations": should be clarified. If you used the SOCAT grided data tracks in your LET experiments, please change to "SOCAT observation-based data" or "SOCAT gridded data".

11. Lines 89-90: "*in an ESM, surface ocean $p$CO$_2$ is known at all times and locations*": not precise enough. It depends on which approximations and computational resources. So far, the models have been derived at 1° or 0.25° and monthly resolutions?

12. Lines 161-162: "*where $pCO_2$ mean and SST mean is the long-term mean of surface ocean $pCO_2$ and temperature, respectively, using all 1°x1° grid cells from the testbed*": $pCO_2$ mean is different regionally, why you don't compute a global map of $pCO_2$ mean?

13. Lines 165-168: Please clarify. The authors have excluded $pCO_2$-Residual which have values below $-250$ $\mu$atm or over 250 $\mu$atm. They mention that such outliers correspond to model values higher than the maximum SOCAT data (816 $\mu$atm) and that do not reflect reality. It is not correct. First, both negative and positive $pCO_2$-Residual values can not represent the upper bound of SOCAT data. Second, SOCAT only covers a tiny portion of the global ocean at a monthly time scale, and there might exist unobserved $pCO_2$ values higher than 816 $\mu$atm (e.g., over permanently or seasonally strong upwelling regions: Eastern Equatorial Pacific, Western Arabian Sea, Benguela, etc).

14. Lines 310-311: "*Our presentation of global maps is limited to runs 'x5_5Y_W' (5022 observations) and 311 'Z_x4_10Y_YR' (7600 observations)*". The information of gridded data used in the experiments should be declared in addition to the number of observations by USVs.

15. Lines 319-321: How did the authors compute Bias (and RMSE) over the global ocean? In order to fairly compare the results of two or more runs (e.g., zigzag vs one-latitude, SOCAT vs SOCAT+USV), error statistics are computed on model-based data excluding all used in ML training. Specifically, the evaluation should not consider 'zigzag+one-latitude' ('SOCAT+USV') $pCO_2$ data.

16. Figures S4 and S5 show cyclic marks (it would be exposed clearly if the authors use a discrete colormap with a low number of colors). Would they be imprints of a driver variable?

17. Figures 5 and 8: The author should report the number of data gridded from USV observations used in ML training. And the error statistics must be computed on the evaluation data (i.e., model-truth-based data excluding all the training data). Figure 8's caption: The mean of RMSEs here is computed with respect to space or time? Instead, the author should compute the mean of squared errors over the global ocean and the periods of interest and then report its square root.

18. Line 386: ''Z_x10_5Y_YR

19. Lines 497-499: "*Although run 'x13_10Y_W' demonstrates the highest reduction in bias out of all runs, the 'zigzag' runs still reduce bias in the Southern Ocean by 44-65 % (vs. 77 % for run 'x13_10Y_W')*". The evaluation should not put high confidence on the bias reduction since this statistic is computed as the mean of negative and positive differences between $pCO_2$ estimates and model truth. Reviewer agrees that the bias can be used to assess model over- or underestimation but RMSD is a better metric for an overall evaluation.

20. Lines 536-541: "*To better understand this discrepancy, we performed an additional experiment based on run 538 'Z_x10_5Y_YR', but assumed sampling every year for the entire testbed period (i.e., 1982-2016). The results from this experiment show a significant eduction in the temporal variability of reconstruction bias; with the additional USV sampling, the reconstructed Southern Ocean air-sea CO2 flux closely matches the 'model truth' for the entire testbed duration (Fig. S14).*". Here biases increases in the last two decades that do not reflect the increase in the number of SOCAT (SOCAT+USV) data as shown in the previous results.

21. Lines 552-554: "*Further, we find that this modest amount of additional Saildrone USV sampling increases the global and Southern Ocean air-sea CO2 flux by up to 0.1 Pg C yr-1, 25% of the uncertainty in the ocean carbon sink*". The increase in global ocean CO2 sink estimated by the LET testbed can not be compared with the uncertainty derived from the GCB's quantification [Friedlingstein et al., 2022]. First, they are two different statistics. Second, the GCB's uncertainty is computed based on the ensemble of different data mapping and modeling methods, and thus the value might be significantly larger than the one estimated by each method itself.

**References**

S. M. Bushinsky, P. Landschützer, C. Rödenbeck, A. R. Gray, D. Baker, M. R. Mazloff, L. Resplandy, K. S. Johnson, and J. L. Sarmiento. Reassessing southern ocean air-sea co2 flux estimates with the addition of biogeochemical float observations. *Global Biogeochemical Cycles*, 33(11):1370–1388, 2019.

A. Denvil-Sommer, M. Gehlen, and M. Vrac. Observation system simulation experiments in the atlantic ocean for enhanced surface ocean $pco_2$ reconstructions. *Ocean Science*, 17(4):1011–1030, 2021. doi: 10.5194/os-17-1011-2021. URL https://os.copernicus.org/articles/17/1011/2021/.

P. Friedlingstein, M. O'Sullivan, M. W. Jones, R. M. Andrew, L. Gregor, J. Hauck, C. Le Quéré, I. T. Luijkx, A. Olsen, G. P. Peters, W. Peters, J. Pongratz, C. Schwingshackl, S. Sitch, J. G. Canadell, P. Ciais, R. B. Jackson, S. R. Alin, R. Alkama, A. Arneth, V. K. Arora, N. R. Bates, M. Becker, N. Bellouin, H. C. Bittig, L. Bopp, F. Chevallier, L. P. Chini, M. Cronin, W. Evans, S. Falk, R. A. Feely, T. Gasser, M. Gehlen, T. Gkritzalis, L. Gloege, G. Grassi, N. Gruber, O. Gürses, I. Harris, M. Hefner, R. A. Houghton, G. C. Hurtt, Y. Iida, T. Ilyina, A. K. Jain, A. Jersild, K. Kadono, E. Kato, D. Kennedy, K. Klein Goldewijk, J. Knauer, J. I. Korsbakken, P. Landschützer, N. Lefèvre, K. Lindsay, J. Liu, Z. Liu, G. Marland, N. Mayot, M. J. McGrath, N. Metzl, N. M. Monacci, D. R. Munro, S.-I. Nakaoka, Y. Niwa, K. O'Brien, T. Ono, P. I. Palmer, N. Pan, D. Pierrot, K. Pocock, B. Poulter, L. Resplandy, E. Robertson, C. Rödenbeck, C. Rodriguez, T. M. Rosan, J. Schwinger, R. Séférian, J. D. Shutler, I. Skjelvan, T. Steinhoff, Q. Sun, A. J. Sutton, C. Sweeney, S. Takao, T. Tanhua, P. P. Tans, X. Tian, H. Tian, B. Tilbrook, H. Tsujino, F. Tubiello, G. R. van der Werf, A. P. Walker, R. Wanninkhof, C. Whitehead, A. Willstrand Wranne, R. Wright, W. Yuan, C. Yue, X. Yue, S. Zaehle, J. Zeng, and B. Zheng. Global carbon budget 2022. *Earth*

*System Science Data*, 14(11):4811–4900, 2022. doi: 10.5194/essd-14-4811-2022. URL https://essd.copernicus.org/articles/14/4811/2022/.

J. Hauck, C. Nissen, P. Landschützer, C. Rödenbeck, S. Bushinsky, and A. Olsen. Sparse observations induce large biases in estimates of the global ocean co2 sink: an ocean model subsampling experiment. *Philosophical Transactions of the Royal Society A*, 381 (2249):20220063, 2023.

P. Landschützer, T. Tanhua, J. Behncke, and L. Keppler. Sailing through the southern seas of air–sea co2 flux uncertainty. *Philosophical Transactions of the Royal Society A: Mathematical, Physical and Engineering Sciences*, 381(2249):20220064, 2023. doi: 10.1098/rsta.2022.0064. URL https://royalsocietypublishing.org/doi/abs/10.1098/rsta.2022.0064.

---

## Author Comment (AC1)

*Response to RC1*

*We would like to thank the reviewer for highly constructive and helpful comments and feedback. We have carefully addressed each question/comment and made changes where we agree that this would improve the manuscript. We especially think the added discussion and figures regarding the testbed spread improved the manuscript. We have provided an itemized list below detailing our responses (in italic font) to the reviewer's suggestions.*

I have some reservations about the way the authors have used their method.

I would expect a more quantitative estimate of the magnitude of the differences potentially detected between the different experiments performed.

The results and discussion of the decadal variability of the oceanic $CO_2$ sink need to be improved (see specific comments).

**Specific comments:**

**1**) To calculate the residual $pCO_2$, the authors used the equation in line 160. In this equation, the term $pCO_2^{mean}$ came from "surface ocean $pCO_2$ […] using all 1°x1° grid cells from the testbed" (line 162). But in the original publication of the methodology (equation 2 in Bennington et al., 2022a), the term $pCO_2^{mean}$ comes from an initial reconstruction of $pCO_2$. Therefore, in this submitted manuscript, by using model outputs instead of an initial reconstruction of $pCO_2$ fields, the authors assumed that their method would be able to perfectly reconstruct the long-term average $pCO_2$ at each grid cell. It seems to me that to obtain a more accurate evaluation of their method (i.e., more accurate observing system simulation experiments) the authors should follow the steps as they were originally published. If this is not possible, could the authors explain why and prove that this assumption does not influence their results.

*The reviewer raises a valid question that the method used here varies slightly in this way from the method presented in Bennington et al. (2022), given the testbed approach utilized here. However,*

*previous work has found that this small difference in methodology does not have a large impact on the result. Bennington et al. (2022) did a sensitivity test of the $pCO_2$-Residual reconstruction to the source of mean $pCO_2$, by experimenting using the Takahashi $pCO_2$ climatology (Takahashi et al., 2009) as well as the mean $pCO_2$ of the SeaFlux observation-based products (Fay et al., 2021; Gregor & Fay, 2021). They found that alternative sources of the initial $pCO_2$ map had little influence on the reconstruction. For this reason, we have chosen to increase the efficiency of our data processing pipeline by using the full model field as our mean $pCO_2$ to calculate $pCO_2$-T and $pCO_2$-Resisudal.*

**2**) To calculate the net sea–air $CO_2$ flux, authors used (line 272): "EN4.2.2 salinity (Good et al., 2013), SST and ice fraction from NOAA Optimum Interpolation Sea Surface Temperature V2 (OISSTv2) (Reynolds et al., 2002), and surface winds and associated wind scaling factor from the European Centre for Medium-Range Weather Forecasts (ECMWF ERA5 sea level pressure (Hersbach et al., 2020)". But as mentioned in line 95: "The goal here is to assess the accuracy with which an ML algorithm can reconstruct the 'model truth'". Therefore, I would expect the model outputs (some of which have already been used for $pCO_2$ reconstruction) to be used to calculate the $CO_2$ flux, rather than observational data which may have different variabilities and/or trends to those simulated. The authors could then compare this calculated $CO_2$ flux to the simulated $CO_2$ flux (and not to a "model truth" $CO_2$ flux from simulated $pCO_2$ fields mixed with observational data). This is particularly important when the authors are discussing the ability of their method to reproduce $CO_2$ flux variability (see my next comment).

*We completely understand the reviewer's point of using model output instead of observational data to calculate flux. However, it is the winds that have the largest impact on flux calculations (Fay et al., 2021), and temporally high-resolution output is not available for the testbed. Only monthly model output is available, and this is not sufficient for the flux calculation due to the square dependency of wind speed. We therefore used the ERA5 wind product, a choice consistent with Gloege et al. (2021) who also used the Large Ensemble Testbed to reconstruct $pCO_2$. Given the necessity to use observed winds, we also use observations for all necessary variables for the flux calculation (Fay et al., 2021), instead of mixing model output and observations.*

*Further, we wish to emphasize that the goal of this project is not to calculate real-world fluxes, but, instead, to better understand how sampling impacts the resulting pCO₂ fields and from pCO₂, the flux. For our study, the most important factor is to calculate consistently for all the experimental runs so that we can make direct comparisons. Therefore, using the same inputs to the flux calculation for each of the three models is also desirable to isolate this comparison. It would certainly be interesting to compare fluxes calculated by different methods (observations vs. model output), however this would be beyond the scope of this paper as we are not evaluating methods of flux calculation, but rather evaluating the impacts of sampling.*

**3**) Authors wrote line 531: "The SOCAT baseline demonstrates a weakening of the global and Southern Ocean carbon sink in the 2000s (Figs. 10, S12), which is in agreement with various data products using real-world SOCAT data". The weakening of the Southern Ocean carbon sink occurred in the 1990s (Le Quéré et al., 2007), while a reinvigoration of the sink was observed during the 2000s (Landschützer et al., 2015). The authors therefore need to revise their text. More importantly, this study focuses on the ability of the authors' method to reproduce "model-truth" variability and not the "real-world" variability. Consequently, I would suggest calculating certain metrics of variability (for example, the size of decadal variability or trends) from simulated $CO_2$ fluxes (and not from recalculated "model-truth" $CO_2$ fluxes, see my previous comment) and comparing the values of these metrics with the values that would be obtained when reconstructed $CO_2$ fluxes are used. Because, otherwise, it assumes that all models perfectly reproduce the variability of the 'real world', which might not be the case.

*We were referring to the distinct "peak" of the weakening of the sink that can be seen around the year 2000, however, we have re-phrased this sentence as suggested by the reviewer:*

*"The 'SOCAT-baseline' demonstrates a weakening of the global and Southern Ocean carbon sink starting in the 1990s with a peak around year 2000 (**Figs. 10, S18**), which is in broad agreement with various data products using real-world SOCAT data (e.g., Gruber et al., 2019; Landschützer et al., 2015; Bushinsky et al., 2019; Bennington et al., 2022; Gloege et al., 2022)".*

*We agree with the reviewer that diving deeper into understanding the flux variability, and comparing fluxes based on the testbed vs. observations would be valuable and we appreciate their*

*suggestion. We believe however that this deserves a more in-depth discussion that will be best presented as an individual paper, and we are planning to explore this further in a future study (this is mentioned in the discussion: "we will further explore this issue in future work"). To avoid a lengthy discussion, we would like to restrict the main focus of this study to assessing the impacts of sampling by using the testbed.*

**4**) The LET has 75 members (i.e., simulations). For each experiment, the values given in the manuscript and in the figures are for the most part averages calculated over the 75 members of the ensemble. But no information is given on the dispersion (or confidence interval) around these averages. It is therefore not possible to assess whether the differences mentioned between the experiments are significant or not.

For example:

- The interpretation of Figure 5 (line 335): "The 'one-latitude' 'high-sampling' run 'x13_10Y_J-A' (44,250 observations) show similar bias or is outperformed by all 'zigzag' runs as well as the 'one-latitude'-runs that restrict sampling to southern hemisphere winter months (i.e., 'x5_5Y_W' and 'x13_10Y_W')." How similar or superior is the performance? Is it true for all members?

- Line 346: "Run 'Z_x10_5Y_W', which has the lowest bias out of the 'zigzag' runs (Fig. 5), shows improvement even further back in time, until the beginning of the testbed period (Fig. S6)." Is it really significant?

I would therefore suggest not only reporting the averages over the 75 members, but also taking advantage of the study of the spread around these averages.

*We thank the reviewer for this suggestion, and in the revised version we have included additional supplementary figures showing the spread amongst ensemble members (**Figs. S8, S10, S14, S16** – these are shown below). Since we are comparing several experiments, it would be difficult to interpret figures showing the spread of 75 members of 10 different experiments, so we chose to keep the figures showing the testbed mean in the main text. It is important to note that in order to fairly compare sampling experiments, it is critical to compare the same ensemble member for each experiment. By that we mean that performance metrics must be calculated based on the same*

member's 'reconstruction vs. truth pair' for each of the 10 sampling experiments. For example, the 'reconstruction vs. truth pair' for CESM member 001 for experiment 1 must be compared to the 'reconstruction vs. truth pair' for CESM member 001 for experiment 2 and so on. There are 75 members in our testbed, and thus, for each experiment, there are 75 'reconstruction vs. truth pairs'. As shown by our supplementary figures (and additional figures below), overall, the mean calculations reflect the majority of individual members in terms of how the different experiments compare to each other.

However, we agree with the reviewer that it is important to show the spread. We have tried to make it more clear throughout the text that we are comparing mean values, but that there is a spread. We added this sentence to **Section 2.3** (Statistical Analysis in the Testbed): "We focus our discussion on the mean across 75 members of the testbed for bias and RMSE. The spread across testbed ensemble members is non-negligible and will be the focus of future work; here, we present the testbed spread primarily in the **Supplement**".

Further, a recent study by Hauck et al. (2023) performed similar sampling experiments, but used a different type of reconstruction method and testbed (i.e., a single hindcast model), and show that additional autonomous sampling leads to a weakened Southern Ocean sink, which is the opposite to our findings. This study was not published when we submitted our initial manuscript, but in the revised version we have added a paragraph to the discussion which touches upon the potential importance of the testbed spread:

"Bushinsky et al. (2019) and Hauck et al. (2023) performed similar sampling experiments as presented here, by comparing ML surface ocean $pCO_2$ reconstructions based on SOCAT vs. additional SOCCOM or ideal virtual floats. These studies showed that SOCAT sampling alone overestimates the $CO_2$ uptake in the Southern Ocean, and that additional floats reduce this overestimation, leading to a decreased (weakened) ocean carbon sink.  In contrast, we find that the $pCO_2$-Residual method underestimates the $CO_2$ uptake with only SOCAT sampling, and that adding USVs increased (strengthened) the Southern Ocean and global ocean sink by up to 0.1 Pg C yr$^{-1}$ (**Figs. 10**, **S18**; **Table S2**).

Going forward, additional studies are needed to better understand why these results suggest a different direction of the sink change with additional sampling. These differences could stem from the use of different reconstruction methods assessed. Hauck et al. (2023) used the MPI-SOM-FFN and CarboScope/Jena-MLS reconstruction methods, while we use the $pCO_2$-Residual method. Another substantial difference between the studies is the models and numbers of ensemble members used as the testbed. Hauck et al. (2023) use a single hindcast model, while we use 25 members each from three Earth System Models. We find substantial spread across these 75 members (**Figs. S8, S10, S14, S16**), indicating that model structure and internal variability significantly impact results. Our study and Hauck et al. (2023) use different approaches for the calculation of fluxes, which could also be a factor. Targeted, coordinated studies using multiple reconstruction approaches with consistent testbed structures and experimental approaches are clearly needed (Rödenbeck et al., 2015). Despite this need for this additional work, studies do agree that additional Southern Ocean observations could significantly improve reconstructions of air-sea $CO_2$ fluxes".

_Answers to the reviewer's specific questions above:_

1. Below, we show the bias (over the Southern Ocean for the period of 2006-2010) of each individual member of the models in the testbed, comparing the high-sampling run 'x13_10Y_J-A' with the equivalent run that restricts sampling to southern hemisphere winter months ('x13_10Y_W'). As shown by the figure below, the majority (~ 80%) of members for run 'x13_10Y_W' (winter sampling) outperform (i.e., have a bias closer to zero) those of run 'x13_10Y_J-A' (Jan-Aug sampling), reflecting the ensemble means shown in **Figure 5**.

[Figure]

*2A. Below, we present zonal annual mean Hovmöller plots showing the change in bias when comparing run 'Z_x10_5Y_W' to the 'SOCAT-baseline'. As shown by the figure below, all models show improvement back in time beyond the additional sampling duration (2012-2016), reflecting the ensemble mean shown in **Figure S6**, but there is less improvement for GFDL members compared to CESM and CanESM2.*

[Figure]

*2B. To examine individual members, we plot time series of bias for run 'Z_x10_5Y_W' and the 'SOCAT-baseline' averaged over the area of highest improvement shown in **Fig. S6** (between 50°S and 35°S). These figures show improvement in bias compared to the 'SOCAT-baseline' already in the beginning of the testbed period for the majority of members, but more so for CESM and CanESM2 compared to GFDL.*

[Figure]

[Figure]

*Overview of new supplementary figures showing the ensemble spread (**S8, S10, S14, S16**):*

[Figure]

*New Fig S8*

[Figure]

CESM ensemble spread

*New Fig S10*

[Figure]

*New Fig S10 cont.*

*New Fig S10 cont.*

[Figure]

**A** Southern Ocean (2006/2012-2016)

[Figure]

**B** CESM - Southern Ocean (2006/2012-2016)

[Figure]

**C** GFDL - Southern Ocean (2006/2012-2016)

[Figure]

**D** CanESM2 - Southern Ocean (2006/2012-2016)

*New Fig S14*

[Figure]

CanESM2 ensemble spread

SOCAT-baseline
Z_x4_10Y_W
Z_x4_10Y_YR
Z_x10_5Y_W
Z_x10_5Y_YR

*New Fig S16*

[Figure]

CESM ensemble spread

GFDL ensemble spread

*New Fig S16 cont.*

*New Fig S16 cont.*

**Technical corrections:**

**5**) Line 37, Please explain the acronym "fCO$_2$". Note that the term "pCO$_2$" is also used in the manuscript. Although understandable to researchers working on this topic, it is less clear to a wider audience. Therefore, authors should be more careful about the terms they use, especially in the Abstract and Introduction sections.

*In the revised version we have defined fCO$_2$, which is the fugacity of carbon dioxide (fCO$_2$) as opposed to pCO$_2$ which is the partial pressure of CO$_2$ in the ocean. The fCO$_2$ is equal to the pCO$_2$ corrected for non-ideality of CO$_2$ solubility in water using the virial equation of state (Weiss 1974). The fugacity correction for surface water is 0.996 and 0.997 at 0 °C and 30 °C respectively (Dickson et al. 2007), or 0.7 to 1.2 μatm lower than the corresponding pCO$_2$, and depends primarily on temperature for the conversion, although pressure is also included in the conversion equation. It is common practice in the observational community to report values as fCO$_2$ as this is what is released in the SOCAT database, but model output is typically reported as pCO$_2$ which is why we have chosen to go with that variable in this study.*

**6**) Line 178, Why aren't the number of decision trees and depth levels different for each reconstruction?

*The depth levels and decision trees are fixed, which we have now stated in the main text. The depth levels and decision trees used represent the optimized parameters for this type of reconstruction. The dominating input for all experiments is based on the SOCAT coverage, and the different USV experiments represent a small increase in the data density. Further, increasing the maximum depth level would make each decision more complex, making the final algorithm less generalizable. Adding more trees is not necessarily going to improve the overall algorithm. Finally, as we are comparing how sampling impacts the reconstruction, changing the decision trees and depth levels for each experiment would make it difficult to assess whether or not potential changes in bias and RMSE are due to the different sampling strategies or the optimization process.*

**7**) Line 188, After reading this sentence, I wasn't sure whether the authors were always going to use the "unseen" values. Could the authors be clearer?

*This should have been communicated more clearly, and we have now revised this sentence and added some more information: "Here, we calculate error statistics based on the full reconstruction ($pCO_2$ from all 1°x1° grid cells of the testbed, except for those masked or filtered out). In the full reconstruction, ~ 99 % of the data do not correspond to SOCAT or Saildrone USV observations used to train the algorithm (**Fig. S1**). Training data would ideally be removed before performance evaluation, but since the training data represent only ~ 1 %, the impact of not removing them is negligible (**Fig. S2**)".*

**8**) Line 203: "2) potential future meridional USV observations ('zigzag' track)". Are they realistic? I found some elements of response later in the text, but it would be good to know here whether all the experiments are realistic or not.

*The reviewer raises an important question, and as pointed out, we touch upon this in **Section '2.4.2 Zigzag runs'** and in the discussion. The potential future meridional USV track has been developed in collaboration with experts from the ocean observing community to test realistic sampling. Due to the USV technology, Saildrones can sample meridional gradients, as opposed to other autonomous platforms. Further, we account for limiting incoming solar radiation to power the Saildrone below 55° S. **Section 2.4** is meant to provide an overview of the different type of experiments we have performed. This section already provides a lot of information, and in order not to exhaust the reader with details, we chose to focus on the details under **Section '2.4.2 Zigzag runs'** instead. However, we added the word "realistic" and refer to further information in **Section 2.4.2**. We also added some more information under section **'2.4.2 Zigzag runs'**: "Saildrone USVs can operate at a speed capable of covering the spatial extent of meridional gradients in the Southern Ocean (Djeutchouang et al., 2022). However, Saildrone USVs are solar powered, and thus their range is restricted by the availability of solar radiation. To account for this and maintain a realistic sampling scenario, sampling occurs only to a maximum latitude of 55° S in these experiments".*

**9**) Table 1: I suggest replacing table 1 with table S1. This is because the information in table 1 is repeated in table S1, and table S1 contains important values that the reader should be able to access easily.

*This has been replaced in the revised manuscript.*

**10**) Line 268, Why not use the same method across all models to calculate $pCO_2^{atm}$? Do all the values obtained take into account the contribution of water vapor pressure?

*The reviewer raises a valid question. The reason for this is that the GFDL model output that we have access to includes the $pCO_2^{atm}$ variable, while for CanESM2 and CESM we do not have this output variable. Therefore, the atmospheric value had to be calculated for these two models. Each individual model defines its own atmosphere concentration, and some models account for water vapor pressure and others do not when running their model. In GFDL and CESM, the contribution of water vapor pressure is taken into account, but this is not the case for CanESM2. Thus, when calculating $pCO_2^{atm}$ for CanESM2 and CESM, the contribution of water vapor pressure was taken into account for only CESM. We now specify that "the contribution of water vapor pressure was corrected for in CESM and GFDL".*

**11**) Line 293: "where algorithm generally overestimates $pCO_2$". This is not the case for the Atlantic sector of the Southern Ocean.

*With this statement we were just trying to convey that, overall, $pCO_2$ is generally overestimated in the Southern Ocean, however, the reviewer is correct that parts of the Atlantic section show an underestimation. We have revised this sentence: "RMSE is highest in the Eastern Tropical and Southeastern Pacific Ocean and in the Southern Ocean, where the algorithm generally overestimates $pCO_2$ (i.e., positive bias;* **Fig. 3a**)*, with some exceptions in the Atlantic section".*

**12**) Figure 3, colour scale: The colour scales need to be harmonised. In panel a, a white colour means a good value, whereas in panel b, it means a bad value.

*We agree with the reviewer, and we tested several different colormaps, however, if we switch the colors in* **Fig. 3** *(i.e., dark color equals "worse"), we would have the same problem in our maps showing our main results (***Figs. 4, 6, 7, 9***). These maps do not show RMSE for each USV experiment, but rather the difference in RMSE between the experiments and the 'SOCAT-baseline'. We could choose a completely different colormap for RMSE in* **Fig. 3***, but for consistency, chose to use the same range of colors for RMSE (and bias) throughout the paper.*

**13**) Figure 3, line 301 to 307: All this information is already present in the text. Please write shorter figure captions. This is a general comment, not just on figure 3.

*Noted, and revised.*

**14**) Line 318 and wherever necessary in the text: "…where the baseline reconstruction…" Please, use the expression "SOCAT baseline" that was introduced in the method section.

*Noted, and revised.*

**15**) Line 384, Please delete the reference to "bias". This was introduced in the previous section.

*Noted, and revised.*

**16**) Line 493, why not excluding the hypothetical data points that would be covered by sea ice?

*The seasonal ice coverage in high latitudes varies, and the sea-ice fraction is uncertain. We chose to show the map of the global sea-ice extent as defined by the SeaFlux product, which is from NOAA OISSTv2 (Reynolds et al., 2002) as an example. Since the sea-ice fraction is uncertain and varies by month, we chose to show where reconstructions could significantly improve regardless of potential ice coverage. If current/future technology allows for sampling in these high-latitude areas it is important to know the extent of the potential improvement.*

**17**) Figure 10: The figure starts in 1985 and not 1982, why?

*The flux calculations begin in 1985 because this corresponds to the earliest SeaFlux inputs. We now add mention of the 1985 start in **Section 2.5**.*

**18**) Figure S3: Because you focused on the open-ocean (line 123), non-open-ocean data should be removed as they were not use for the training, is it right? Does this drastically modified the data availability and explain why better results are obtained from 1990?

*Testbed output for coastal areas, the Arctic Ocean and marginal seas were removed before training in all experimental runs, and also when comparing the experiments to the testbed truth when calculating bias, RMSE and air-sea flux. As shown in **Figs. 3, 4** and **7** (and equivalent figures*

*in the supplement) the white areas represent areas of no data as this was removed. Better results are likely obtained from 1990 because, as shown by **Fig. S3** (**Fig. S5c** in the revised version), SOCAT observations start to drastically increase from these times. This was mentioned in the manuscript: "Considering the change in bias from year-to-year, the 'SOCAT-baseline' shows positive bias at all latitudes in the beginning of the testbed period, before improvement occurs around 1990 (**Fig. 6a**). This is consistent with increasing SOCAT sampling with time for the period considered here (i.e., up to 2016; **Fig. S5c**)".*

**References**

*Bennington, V., Galjanic, T., and McKinley, G. A.: Explicit Physical Knowledge in Machine Learning for Ocean Carbon Flux Reconstruction: The pCO$_2$-Residual Method, Journal of Advances in Modeling Earth Systems, 14(10), https://doi.org/10.1029/2021ms002960, 2022.*

*Dickson, A. G., Sabine, C. L., & Christian, J. R. (Eds): Guide to best practices for ocean CO2 measurement, Sidney, British Columbia, North Pacific Marine Science Organization, 191 (PICES Special Publication 3; IOCCP Report 8), http://dx.doi.org/10.25607/OBP-1342, 2007.*

*Fay, A. R., Gregor, L., Landschützer, P., McKinley, G. A., Gruber, N., Gehlen, M., Iida, Y., Laruelle, G. G., Rödenbeck, C., Roobaert, A., and Zeng, J.: SeaFlux: harmonization of air–sea CO$_2$ fluxes from surface pCO$_2$ data products using a standardized approach, Earth Syst. Sci. Data, 13, 4693–4710, https://doi.org/10.5194/essd-13-4693-2021, 2021.*

*Gloege, L., McKinley, G. A., Landschützer, P., Fay, A. R., Frolicher, T. L., and Fyfe, J. C.: Quantifying Errors in Observationally Based Estimates of Ocean Carbon Sink Variability, Global Biogeochemical Cycles, 35(4), https://doi.org/10.1029/2020gb006788, 2021.*

*Gregor L., & Fay, A. R.: SeaFlux data set: harmonised sea-air CO$_2$ fluxes from surface pCO$_2$ data products using a standardised approach (2021.04, Data set: Zenodo. https://doi.org/10.5281/zenodo.5148460, 2021).*

*Hauck, J., Nissen, C., Landschützer, P., Rödenbeck, C., Bushinsky, S., and Olsen, A.: Sparse observations induce large biases in estimates of the global ocean CO2 sink: and ocean model subsampling experiment, Philosophical Transactions Of the Royal Society A, 381:20220063, https://doi.org/10.1098/rsta.2022.0063, 2023.*

*Takahashi, T., Sutherland, S.C., Wanninkhof, R., Sweeney, C., Feely, R.A., Chipman, D.W., Hales, B., Friederich, G., Chavez, F., Sabine, C. and Watson, A.: Climatological mean and decadal change in surface ocean pCO2, and net sea–air CO2 flux over the global oceans. Deep Sea Research Part II: Topical Studies in Oceanography, 56(8-10), pp.554-577, 2009.*

*Weiss., R.: Carbon dioxide in water and seawater: the solubility of non-ideal gas, Marine Chemistry, 2(3), 203-215, https://doi.org/10.1016/0304-4203(74)90015-2, 1974.*

---

## Author Comment (AC2)

***Response to RC2***

*We would like to express our gratitude to the reviewer for constructive and helpful comments/feedback. We have carefully addressed each question/comment and made changes where we agree that this would improve the manuscript. We have provided an itemized list below detailing our responses (in italic font) to the reviewer's suggestions.*

Despite appreciating the author's efforts in this study, Reviewer has not been convinced by its originality. Based on ESM output, numerous existing research works have shown additional data sampling (e.g., bgcArgo, SOCCOM, Sailboat,...) critical for error reduction in pCO2 and flux estimation over the Southern Ocean and/or the global ocean [Bushinsky et al., 2019, Denvil-Sommer et al., 2021, Hauck et al., 2023, Landschützer et al., 2023]. One suggestion that would add value to the manuscript's findings is an analysis of spatial and temporal variations of flux estimates: to what extent their variability changes subject to the additional data. Some other major concerns are listed below.

*Our study presents new findings that provide more insight into the number of additional samples and spatial pattern, consistent with current technology, that could reduce uncertainty in the ocean carbon sink, particularly in the Southern Ocean. There is no other study quantifying the impacts of meridional sampling by comparing different USV sampling tracks (also taking winter vs. summer sampling into account) in the Southern Ocean by using a Large Ensemble Testbed. Bushinsky et al. (2019) base their experiments on real-world SOCCOM float observations and use the SOM-FFN product for reconstruction. This is an important contribution. However, float-based estimates of $pCO_2$ are not incorporated into the SOCAT database and there are concerns about bias. It is therefore important to test the impact of realistic USV sampling, that can take direct $pCO_2$ observations with low uncertainties, can cover meridional gradients in the Southern Ocean, and are already incorporated into the SOCAT database.*

*The study by Hauck et al. (2023) uses GOBM output from one single model and reconstructs using two reconstruction methods (SOM-FFN and CarboScope), while we use ESM output from 75 different members and the $pCO_2$-Residual method. We also test a very different sampling pattern*

*compared to the "idealized" sampling in Hauck et al. (2023). We do find the study by Hauck et al. (2023) interesting, but note that it was not published when we submitted our initial manuscript. In the revised version we have added a paragraph discussing this study and comparing their results to ours. A key point made is that both Bushinsky et al. (2019) and Hauck et al. (2023) show an overestimation of the ocean sink with current sampling, while we show the opposite – an underestimation of the ocean sink. This further suggests that our study complements previous studies and adds value to this pertinent topic of ocean carbon research. It is important to present studies with different types of testbeds and reconstruction methods, so that we can better understand the impact of adding autonomous observations.*

*The study by Denvil-Sommer et al. (2021) is different to ours as it assesses sampling in the Atlantic Ocean, whereas our study focuses on sampling in the Southern Ocean and we show global reconstructions. Further, their study uses a different reconstruction method and assumes sampling from floats, not USVs.*

Lines 149-153: "To build reconstruction algorithms through the data-driven training that occurs in ML, the statistics in all other algorithms developed to date must identify a function that disentangles these competing effects of SST on pCO2. Here, the algorithm is assisted by removing this known temperature effect, and it must therefore only learn the pCO2 impacts from biogeochemical drivers": there exist many other ML approaches [Friedlingstein et al., 2022] which do not separate the SSTeffects from others on pCO2 but succeeds in estimate pCO2. The major concerns are how to assess the uncertainty derived from SST effect removal and impacts on the experiment outputs.

*Our study is not an evaluation of different ML approaches, but rather an assessment of how sampling impacts $pCO_2$ reconstructions. An evaluation of the method itself has already been performed by Bennington et al. (2022). They demonstrated that the $pCO_2$-Residual method performs better compared to other products when evaluating against independent data. They also showed improved skill when using $pCO_2$-Residual as the target variable as opposed to $pCO_2$. We want to assess how different sampling patterns affect the $pCO_2$ reconstruction. As we use the same*

*method for all experiments, we can directly compare them and evaluate how sampling impacts the reconstructions.*

2. Figure 3: Relatively small bias and RMSE values have shown their imprints on the SOCAT track compared to "unseen" model truth. This evidences the problems of model overfitting. The authors can double-check whether model overfitting comes from the cross-validation technique or the pCO2-Residual method. As the key findings of this manuscript are based on the data reconstruction results, Reviewer suggests the authors to carefully verify their methods and solve the problems of model overfitting before further consideration for publication.

*We would argue that the global mean bias and RMSE for the SOCAT reconstruction is comparable to values shown for $pCO_2$ reconstructions using other methods (e.g., Stamell et al., 2020; Gregor et al., 2019). For example, as shown in* **Figure 3**, *bias generally ranges between -10 to +10 μatm, which is comparable to the study by Hauck et al. (2023). However, after carefully evaluating our calculations following the reviewer's feedback, we noticed an error in our code that calculates the RMSEs. After fixing this error, the mean RMSE values increased by ~ 3-4 μatm.*

**Editorial and specific comments:**

1. Lines 11-12: "anthropogenic" can be removed. The SO has taken up atmospheric CO2 without specifying natural or anthropogenic sources.

*The Southern Ocean actively cycles natural and absorbs anthropogenic carbon. Gruber et al. (2009) demonstrate that the Southern Ocean is a source for natural carbon. The ocean sink for anthropogenic carbon is what we wish to focus on in this discussion.*

2. Line 37: "fCO2" is not defined. "uncertainty of < 5 μatm": this holds only for the measurements chosen to provide gridded SOCAT datasets.

*Noted and revised: "The Surface Ocean $CO_2$ ATlas (SOCAT; Bakker et al., 2016) is the largest global database of surface ocean $CO_2$ observations, with data starting in 1957. The main synthesis*

*and gridded products contain over 33 million high-quality direct shipboard measurements of fCO₂*
*(fugacity of CO₂) with an uncertainty of < 5 µatm (Bakker et al., 2022)".*

3. Line 42: "Observation-based data products" −→ "Data mapping methods".

*We wish to use the term 'observation-based data products' consistently following recent literature*
*(e.g., Fay et al., 2021; Crisp et al., 2022; Friedlingstein et al., 2023).*

4. Line 45: "These data products" −→ "These methods".

*See above comment.*

5. Lines 46-47: please remove or change ";" in the brackets to facilitate reading. You can use "-"
instead. Line 47: "xCO2; atmospheric CO2" −→ "atmospheric CO2 - xCO2"

*Noted and revised.*

6. Line 48: "where these are co-located" −→ "where their available data are colocated".

*We chose to keep the original sentence.*

7. Lines 50-51: "Since the data products rely on observations to train the algorithms and thus
produce these relationships": please rephrase this sentence. Data products do not train algorithms
and produce relationships, but the ML-based methods themselves estimate the function between
predictors and target data!

*Noted and revised: "Since the data products rely on pCO₂ observations to estimate functions*
*between the target and driver variables, data sparsity remains a fundamental limitation to this*
*technique".*

8. Line 57: "indirect pCO2 estimates": can you define this term? Are they computed from float measurements of other carbonate variables?

*Noted and revised. We added this sentence: "These large uncertainties and biases arise when $pCO_2$ is not measured directly as in the observations included in SOCAT, but is rather estimated using measurements of pH combined with a regression-derived alkalinity estimate (Williams et al., 2017; Gray et al., 2018). SOCAT includes only direct $pCO_2$ observations".*

9. Lines 67-68: "Such improvements in sampling are critically important in the undersampled Southern Ocean": USVs with low measurement uncertainty would prompt to be employed for observing network systems of pCO2 but to draw this statement, it requires to provide the availability of USVs to sample pCO2 by showing the sampling frequency and data coverage area over the SO?

*Additional high-accuracy observations from the sparsely sampled Southern Ocean, such that can be obtained by USVs, are key to provide further constraints on the ocean carbon sink and air-sea flux. We do not believe it is necessary to go into detail about the data coverage over the Southern Ocean, as we reference studies such as Bakker et al. (2016, 2022) describing the SOCAT coverage (which includes the Saildrone observations from Sutton et al. (2021) in the latest version). We also mention that the SOCAT coverage is shown in supplementary **Fig. S3 (Fig. S5** in the revised version).*

10. Line 86: "actual observations": should be clarified. If you used the SOCAT grided data tracks in your LET experiments, please change to "SOCAT observation-based data" or "SOCAT gridded data".

*We have revised the sentence: "However, instead of using real-world observations, we sample the target (i.e., surface ocean $pCO_2$) and driver variables (i.e., SST, SSS, MLD, Chl-a and $xCO_2$) from our Large Ensemble Testbed (LET) of Earth System Models (ESMs) (e.g., Stamell et al., 2020; Gloege et al., 2021; Bennington et al., 2022a)".*

11. Lines 89-90: "in an ESM, surface ocean pCO2 is known at all times and locations": not precise enough. It depends on which approximations and computational resources. So far, the models have been derived at 1 ° or 0.25° and monthly resolutions?

*We are just aiming to convey that an ESM will not have huge gaps like in the real ocean. We have revised the sentence: "First, in an ESM, the surface ocean $pCO_2$ field is provided precisely at all model times and 1°x1° points". The models used in our study have a 1°x1° resolution, which is stated multiple times throughout the manuscript.*

12. Lines 161-162: "where pCO2 mean and SST mean is the long-term mean of surface ocean pCO2 and temperature, respectively, using all 1°x1° grid cells from the testbed": pCO2 mean is different regionally, why you don't compute a global map of pCO2 mean?

*We do compute a mean of $pCO_2$ globally, which is the $pCO_2^{mean}$ and this is used to calculate the residual.*

13. Lines 165-168: Please clarify. The authors have excluded pCO2-Residual which have values below −250 µatm or over 250 µatm. They mention that such outliers correspond to model values higher than the maximum SOCAT data (816 µatm) and that do not reflect reality. It is not correct. First, both negative and positive pCO2- Residual values cannot represent the upper bound of SOCAT data. Second, SOCAT only covers a tiny portion of the global ocean at a monthly time scale, and there might exist unobserved pCO2 values higher than 816 µatm (e.g., over permanently or seasonally strong upwelling regions: Eastern Equatorial Pacific, Western Arabian Sea, Benguela, etc).

*We are not saying that both negative and positive $pCO_2$-Residual values represent the upper bound of SOCAT data. Our statement is "These $pCO_2$-Residual values **generally** correspond to high $pCO_2$, above the maximum value in SOCAT (816 µatm)". By this we mean that the majority of the $pCO_2$-Residual values that have been filtered out represent $pCO_2$ values that are larger than 816 µatm. However, since this seemed to be unclear, we have re-phrased this sentence: "Prior to algorithm processing, $pCO_2$-Residual values > 250 µatm and < -250 µatm from the testbed were*

*filtered out targeting values that are not representative of the real ocean. The majority of the pCO₂-Residual values that were filtered out correspond to high $pCO_2$, above the maximum value in SOCAT (816 µatm; Stamell et al., 2020)".*

14. Lines 310-311: "Our presentation of global maps is limited to runs 'x5_5Y_W' (5022 observations) and 311 'Z_x4_10Y_YR' (7600 observations)". The information of gridded data used in the experiments should be declared in addition to the number of observations by USVs.

*We revised the sentence: "Our presentation of global maps is limited to runs 'x5_5Y_W' (5,022 monthly 1°x1° observations) and 'Z_x4_10Y_YR' (7,600 monthly 1°x1° observations)".*

15. Lines 319-321: How did the authors compute Bias (and RMSE) over the global ocean? In order to fairly compare the results of two or more runs (e.g., zigzag vs one-latitude, SOCAT vs SOCAT+USV), error statistics are computed on modelbased data excluding all used in ML training. Specifically, the evaluation should not consider 'zigzag+one-latitude' ('SOCAT+USV') pCO2 data.

*The reviewer is correct - the training data should ideally be removed before computing error statistics. When using actual observations, one would evaluate the reconstruction based on the test set alone. However, since we are using a model testbed, we have the opportunity to evaluate against $pCO_2$ values from "unseen" grid cells as well. In our study, we compute error statistics based on the full reconstruction, however this should have been communicated more clearly. The training data represents only about 1% of the full reconstruction. Below, we show the 75-member testbed spread in bias and RMSE calculated based on the full reconstruction (what we present in our study) vs. 'unseen' grid cells for the 'SOCAT-baseline'. The difference in mean bias and RMSE between the full and 'unseen' reconstruction is only 0.01 µatm and 0.08 µatm, respectively. The results from the different runs can therefore be compared even though the full reconstruction is taken into account. We agree however with the reviewer that the training data should have been removed. Considering that we would have to re-run all experiments, and it would not change the error statistics significantly or change our conclusions, we chose not to move forward with this for*

*this study. However, for future studies using the testbed, the training set will be removed before calculating statistical metrics.*

*We now add mention of this: "Here, we calculate error statistics based on the full reconstruction ($pCO_2$ from all 1°x1° grid cells of the testbed, except for those masked or filtered out). In the full reconstruction, ~ 99 % of the data do not correspond to SOCAT or Saildrone USV observations used to train the algorithm (**Fig. S1**). Training data would ideally be removed before performance evaluation, but since the training data represent only ~ 1 %, the impact of not removing them is negligible (**Fig. S2**)". (**Figs. S1** and **S2** are shown below).*

[Figure]

***Figure S1**: Maps of the full pCO₂-Residual reconstruction (all 1°x1° grid cells of the testbed, except for those masked or filtered out; see **Section 2.1** and **2.2**), 'unseen' reconstruction (all 1°x1° grid cells that do not correspond to SOCAT observations), and training data from the testbed. The maps show data from CESM member 001 for the month of March 2016 for the 'SOCAT-baseline'.*

*Numbers on panels represent the total monthly 1°x1° grid cells for the entire testbed period (1982-2016) for each group of data.*

[Figure]

***Figure S2****: Spread of bias (**a**) and RMSE (**b**) for the 75 members of the Large Ensemble Testbed for the 'unseen' and full reconstruction for the 'SOCAT-baseline'. The 'unseen' reconstruction represents independent data, i.e., all 1°x1° grid cells that do not correspond to SOCAT or Saildrone USV observations, and is not part of the training set.*

16. Figures S4 and S5 show cyclic marks (it would be exposed clearly if the authors use a discrete colormap with a low number of colors). Would they be imprints of a driver variable?

*These "cyclic marks" are likely imprints of the three-component n-vector that replaces the longitude and latitude coordinates to continuous values between 0 and 1 (i.e., to avoid the algorithm interpreting 0 and 360 degrees to be far apart; see figure below).*

[Figure]

*Bennington et al. (2022) present global maps (their Fig. 4) of the feature importance of various driver variables used in the surface ocean pCO₂ reconstruction (MLD, SST, Chl-a, location and day of year). Such "cyclic marks" are apparent for "geographic location" and "day of year", but none of the other drivers. We did two test runs (using only one member from the testbed), removing day of year (DOY) and geographic location (n-vector; A, B and C) as inputs for the reconstruction. As shown by the figure below, the "cyclic" marks disappear when the n-vector is removed. When removing the n-vector transformation, however, the reconstruction shows significantly higher bias in the Southern Ocean, so we chose to keep these driver variables.*

[Figure]

17. Figures 5 and 8: The author should report the number of data gridded from USV observations used in ML training. And the error statistics must be computed on the evaluation data (i.e., model-truth-based data excluding all the training data). Figure 8's caption: The mean of RMSEs here is computed with respect to space or time? Instead, the author should compute the mean of squared errors over the global ocean and the periods of interest and then report its square root.

*The number of monthly 1°x1° observations for each experiment is described in **Table 1** as well as shown on the x-axis of **Figure 5** and **8**. This was specified in the **Table 1** caption, but we now specify this in the figure captions as well: "'# additional observations' = number of monthly 1°x1° USV observations in addition to SOCAT". We state in the manuscript that: "The test and validation set each account for 20 % of the data, leaving 60 % for training". For both **Fig. 5** and **8**, the mean is computed with respect to both space (top figure shows global and bottom figure shows Southern Ocean, which in our study is defined as south of 35° S) and time, which is 2006-2016 (for the 10-year sampling) and 2012-2016 (for the five-year sampling). This is stated in the figure headlines.*

*Regarding comment about error statistics, please see answer #15.*

18. Line 386: ''Z_x10_5Y_YR

*Noted and revised.*

19. Lines 497-499: "Although run 'x13_10Y_W' demonstrates the highest reduction in bias out of all runs, the 'zigzag' runs still reduce bias in the Southern Ocean by 44-65 % (vs. 77 % for run 'x13_10Y_W')". The evaluation should not put high confidence on the bias reduction since this statistic is computed as the mean of negative and positive differences between pCO2 estimates and model truth. Reviewer agrees that the bias can be used to assess model over- or underestimation but RMSD is a better metric for an overall evaluation.

*We agree with the reviewer, and that is why we report both bias and RMSE. Our conclusions do not fully rely on bias alone, as is shown throughout the paper. For example, we conclude that the zigzag-runs perform best overall, even though run 'x13_10Y_W' demonstrates a higher reduction with respect to mean bias.*

20. Lines 536-541: "To better understand this discrepancy, we performed an additional experiment based on run 538 'Z_x10_5Y_YR', but assumed sampling every year for the entire testbed period (i.e., 1982-2016). The results from this experiment show a significant eduction in the temporal

variability of reconstruction bias; with the additional USV sampling, the reconstructed Southern Ocean air-sea CO2 flux closely matches the 'model truth' for the entire testbed duration (Fig. S14).". Here biases increases in the last two decades that do not reflect the increase in the number of SOCAT (SOCAT+USV) data as shown in the previous results.

*As shown by the figure below, run 'Z_x10_5Y_YR' (shown in **Fig. 6** in main text) and 'Z_x10_35Y_YR' (shown in **Fig. S14** in supplement; in the revised version, this is now **Fig. S20**) show similar variability the last five years when the sampling is identical. For run Z_x10_5Y_YR', USV observations have been added only for the last five years of the testbed, while for run 'Z_x10_35Y_YR', USV observations have been added for the whole testbed period (35 years). The bias decreases more significantly in the earlier decades for run 'Z_x10_5Y_YR' because there are no additional USV observations at this time, and there are significantly less SOCAT observations in this period compared from 1990 and onwards (see **Fig. S3c**; in the revised version, this is now **Fig. S5c**).*

[Figure]

21. Lines 552-554: "Further, we find that this modest amount of additional Saildrone USV sampling increases the global and Southern Ocean air-sea CO2 flux by up to 0.1 Pg C yr-1, 25% of the uncertainty in the ocean carbon sink ". The increase in global ocean CO2 sink estimated by the LET testbed can not be compared with the uncertainty derived from the GCB's quantification [Friedlingstein et al., 2022]. First, they are two different statistics. Second, the GCB's uncertainty is computed based on the ensemble of different data mapping and modeling methods, and thus the value might be significantly larger than the one estimated by each method itself.

*These values can be compared as they are in the same units. We wish to demonstrate that 0.1 Pg C/yr is a significant reduction. Following the reviewer's comment, we revised the sentence: "Further, we find that this modest amount of additional Saildrone USV sampling increases the global and Southern Ocean air-sea $CO_2$ flux by up to 0.1 Pg C $yr^{-1}$, a quantity equivalent to 25 % of the uncertainty in the ocean carbon sink".*

---

## Referee Report (RR1)

I appreciate the authors' efforts for clarifying the reviewers' concerns and refining the manuscript. This study will be reconsidered for publication as soon as the **remaining issues** are fully addressed.

General comment:

The point is not to highlight the use of any specific type of pCO2 measurements over the others for the estimation of global maps of pCO2. For instance, float-based data provides indirect observations of pCO2 and thus high uncertainty for pCO2 estimates. However, the suggestions learned from the previous works [Bushinsky et al. (2019), Denvil-Sommer et al., 2021, Djeutchouang et al., 2022, Hauck et al., 2023, Landschützer et al., 2023] are to obtain more accurate (precise) estimates of pCO2 by extending the observing systems or considering additional data sources available in space and time. Besides, many of the existing works have exploited the sensitivity of pCO2 and flux estimates to the data sparsity over the Southern Ocean.

However, I agree that Thea Hatlen Heimdal et al have contributed a new finding about different USV sampling strategies to the global reconstruction of pCO2. It's worth to add few sentences in the last paragraph in Section Introduction to bold the new contributions as complements to the previous works. A summary of Section Methods would be enough: e.g. one-latitudes and zigzag sampling, … which differ from the SOCAT+SOCCOM or Argo-float ideal sampling over the global ocean by Hauck et al., 2023).

Specific comments:

I do not support the following arguments of the authors in their responses to the reviewers:

"*We do find the study by Hauck et al. (2023) interesting, but note that it was not published when we submitted our initial manuscript. In the revised version we have added a paragraph discussing this study and comparing their results to ours (lines 933-954). A key point made is that both Bushinsky et al. (2019) and Hauck et al. (2023) show an overestimation of the ocean sink with current sampling, while we show the opposite – an underestimation of the ocean sink.*"

First, I am not aware whether the initial manuscript was submitted to other journals or not. But as tracking the MS record in Biogeosciences, this study first appeared for review in September 2023 while Hauck et al. (2023) was published in March 2023. Second, it has a level of confidence of an overestimation of pCO2 based on present-SOCAT sampling as tested by Bushinsky et al. (2019) and Hauck et al. (2023). pCO2 generally increases over time and mapping methods tend to underestimate pCO2 (thence overestimate fluxes) based on sparse training datasets which have not covered the full range of realistic pCO2 values (many regions with high pCO2 values are unobserved). It's questioning about the distinction between the results in this study and the previous.

Lines 846-854 (in the manuscript with track changes) will need to be revised (or excluded). For such sensitivity tests, one would not expect to see the comparison in performance of different mapping methods but of a fixed method to different sampling scenarios. Even in this study, an ensemble of model output or the methods based on SST-removal effects from pCO2 would add more uncertainty to statistics such as bias, RMSD,... That's why I have suggested analyzing further differences in fluxes' variability (trends, seasonal cycles,...) with respect to different sampling strategies.

The discontinuity in Figures 3 and 7 still persists: we have obviously seen the gradients in RMSD at (SOCAT or zigzag) sampling tracks versus the "unobserved" areas. Therefore, I expect the authors to verify whether their mapping method put much higher weights on sampled locations than "unobserved" regions ('overfitting': i.e. over-exploitation of the entire available data for model training). From a statistical point of view, different mapping methods learned on different model testbeds (i.e. different training data have different data ranges) probably result in different magnitudes of RMSE or Bias. It is not convincing to mention that their mapping method has error values in line with those in the previous study.

Again, in the following sentence and others in the text, please be careful using the phrase "Observation-based data products". Precisely, "mapping methods" have been developed to estimate pCO2 and generate global "Observation-based data products".

Lines 50-52 (in the manuscript with track changes):

"Observation-based data products have been developed to estimate full-coverage surface ocean pCO2 across space and time by extrapolating to global coverage from these sparse SOCAT observations."

---

## Author Response (AR3)

*We would like to express our gratitude to both reviewers and the editor for the second reviews of this manuscript. We have thoroughly addressed all specific comments below. Line numbers refer to the "tracked changes" version of the manuscript. Changes from the previous round of revisions are marked in red. Changes for this second round of revisions are highlighted in yellow. The main changes to the manuscript are:*

- *We included a sentence regarding the source of mean $pCO_2$ for the $pCO_2$-Residual calculation (lines 217-219).*
- *We added a statement explaining why we use observations (and not testbed output) to calculate fluxes (lines 384-388).*
- *In the introduction, we now highlight the new knowledge contributed in our study and how our work complements previous studies (lines 131-133 and 135-140)*
- *We mention the additional importance of sampling masks when comparing testbed studies reconstructing surface ocean $pCO_2$ (line 879 and 212).*
- *We have replaced "observation-based data products" with "mapping methods" throughout the manuscript (line 50 and 209).*
- *We discussion overfitting in the main text (lines 417-423) and in the Supplementary Material (**Supplementary Text A**) and added a supplementary figure (**Fig S6**).*
- *We added a statement which explains that reconstructions using $pCO_2$-Residual as the target variable as opposed to $pCO_2$ leads to higher skill (lower RMSEs) (lines 208-211).*

**Response to Reviewer 1**

I would like to point out that the responses to my comments could have been done better. In my first comment on the changes to the methodology, the authors replied that the required sensitivity analysis had been carried out as part of an earlier publication. But the authors didn't mention any value in their response to my comment and wrote "had little influence on the reconstruction", leaving me not knowing how "little" that is.

*We are sorry that the reviewer feels our responses were insufficient. Our answer refers directly to experiments performed by Bennington et al. (2022), and reported in that publication. Here is the full text in the last paragraph of section 2.3 of that paper. "We tested the sensitivity of the reconstruction to the source of mean $pCO_2$ used in the calculation of $pCO_2$-T with Equation 2, which is then input to the $pCO_2$-Residual calculation in Equation 3. Reconstructions using the Lamont-Doherty Earth Observatory (LDEO) $pCO_2$ climatology (Takahashi et al., 2009) and the mean $pCO_2$ of the SeaFlux observation-based products (Fay et al., 2021). The alternative sources of mean $pCO_2$ did not significantly impact reconstructed $pCO_2$ or resulting air-sea $CO_2$ exchange, so we maintain our own method for the initial reconstruction of $pCO_2$." To be more precise on "little influence" from this work - the test statistics, as reported in Table 3 of Bennington et al. (2022), were not sensitive to the choice of initial $pCO_2$ field.*

*Furthermore, Bennington et al (2022) have already demonstrated the skill of the $pCO_2$-Residual approach using real-world data in their comparisons to independent data at BATS and HOT, and from GLODAP and from the LDEO $pCO_2$ database (i.e., points not in SOCAT). Together, Bennington et al. (2022) provide evidence that this approach performs better, admittedly*

*marginally, compared to other observation-based products. Thus, to use the approach in this Large Ensemble Testbed study focused on the impact of sampling distribution is reasonable without further sensitivity studies on the method itself.*

*In the revised manuscript we added a sentence stating that alternative sources of mean $pCO_2$ have been assessed by Bennington et al. (2022) and that they did not significantly impact the test statistics or reconstructed $pCO_2$ (lines 217-219):*

*"Alternative sources of mean $pCO_2$ were assessed by Bennington et al. (2022a), but they found no significant impact on the test statistics or reconstructed $pCO_2$."*

In the reply to my second comment, the authors wrote "high temporal resolution output is not available for the test bench". Why would the authors need high-resolution model outputs, when the pCO2 reconstruction performed by the method is carried out with the same spatio-temporal resolution as the model outputs (1°x1°, monthly temporal resolution). This argument seems irrelevant, whereas the second part of their reply, which mentions that their aim was not to calculate the real-world fluxes, is more understandable.

*We are sorry that this was unclear. Our goal was to convey the impact on our ability to calculate air-sea $CO_2$ fluxes given the lack of temporally high-resolution output of winds. Because of the square dependence of the flux on winds, one needs high-resolution (3 or 6 hourly) winds to calculate fluxes. Since only monthly model output for the winds is available, we cannot use model-based winds for the flux calculation.*

*We add to the revised manuscript a statement that we do not have high-resolution output of **winds** (lines 384-388):*

*"Winds have the largest impact on flux calculations (Fay et al., 2021), and temporally high-resolution output is not available for the LET. Monthly output is available, but this is not sufficient for the flux calculation due to the square dependency of wind speed (Wanninkhof, 2014). Given the necessity to use observed winds, for consistency, we use observations for all necessary variables for the flux calculation."*

Finally, in their response to my 4th specific comment, the authors wrote three paragraphs that were primarily aimed at the second reviewer's comments. These paragraphs did not address my comment and appear to have been poorly copied and pasted. Therefore, additional care needs to be taken.

*We are sorry that the reviewer does not feel like we appropriately responded to this valuable comment that helped us to significantly improve the manuscript. It led us to include testbed spread comprehensively across the manuscript. The first 2 paragraphs in the previous response do directly address Reviewer 1's comments. The following three paragraphs were, indeed, part of an answer to reviewer 2. We believe these amplify the reviewer's point about the necessity of showing the testbed spread. We are sorry Reviewer 1 finds these additional paragraphs unnecessary.*

**Response to Reviewer 2**

General comment:

The point is not to highlight the use of any specific type of pCO2 measurements over the others for the estimation of global maps of pCO2. For instance, float-based data provides indirect observations of pCO2 and thus high uncertainty for pCO2 estimates. However, the suggestions learned from the previous works [Bushinsky et al. (2019), Denvil-Sommer et al., 2021, Djeutchouang et al., 2022, Hauck et al., 2023, Landschützer et al., 2023] are to obtain more accurate (precise) estimates of pCO2 by extending the observing systems or considering additional data sources available in space and time. Besides, many of the existing works have exploited the sensitivity of pCO2 and flux estimates to the data sparsity over the Southern Ocean. However, I agree that Thea Hatlen Heimdal et al have contributed a new finding about different USV sampling strategies to the global reconstruction of pCO2. It's worth to add few sentences in the last paragraph in Section Introduction to bold the new contributions as complements to the previous works. A summary of Section Methods would be enough: e.g. one-latitudes and zigzag sampling, … which differ from the SOCAT+SOCCOM or Argo-float ideal sampling over the global ocean by Hauck et al., 2023).

*Thank you for this clarification. We have added some sentences in the last paragraph in the introduction to highlight how our work complements these previous studies (lines 131-133 and 135-140):*

*"We test the impact of two different USV Southern Ocean sampling schemes, the first based on a sampling campaign completed in 2019 (Sutton et al., 2021), and the second on logistically feasible potential future meridional sampling."*

*"Combined, the sampling patterns tested here complements previous studies exploring the impact of additional sampling in the Southern Ocean based on idealized full global coverage of floats, and float observations from recent deployments, including the Southern Ocean Carbon and Climate Observations and Modeling (SOCCOM) project, moorings and sailboats (Bushinsky et al., 2019; Denvil-Sommer et al., 2021; Djeutchouang et al., 2022; Hauck et al., 2023; Behncke et al., 2024; Landschützer et al., 2023)."*

Specific comments:

I do not support the following arguments of the authors in their responses to the reviewers:

"We do find the study by Hauck et al. (2023) interesting, but note that it was not published when we submitted our initial manuscript. In the revised version we have added a paragraph discussing this study and comparing their results to ours (lines 933-954). A key point made is that both Bushinsky et al. (2019) and Hauck et al. (2023) show an overestimation of the ocean sink with current sampling, while we show the opposite – an underestimation of the ocean sink."

First, I am not aware whether the initial manuscript was submitted to other journals or not. But as tracking the MS record in Biogeosciences, this study first appeared for review in September 2023 while Hauck et al. (2023) was published in March 2023.

*We apologize for our oversight here. We should have said in our first response to reviewers that we were not aware of this publication when we submitted the first version of this manuscript.*

Second, it has a level of confidence of an overestimation of pCO2 based on present-SOCAT sampling as tested by Bushinsky et al. (2019) and Hauck et al. (2023).

*We do not understand this comment. We cite directly the overestimation with SOCAT-only that has been reported by Bushinsky et al. (2019) and Hauck et al. (2023) in comparing to SOCAT+float sampling (and compared to the model truth):*

- *Bushinsky et al. (2019): "The combined SOCAT+SOCCOM product yields a Southern Ocean sink that is 0.4 Pg C/yr weaker over 2015-2017 than that calculated from shipboard data alone" (page 1385, Section 3.4).*
- *Bushinsky et al. (2019): "...the SOCAT-only uptake 0.22 Pg C/yr stronger than the model and the SOCAT+SOCCOM uptake 0.14 Pg C/yr stronger than the true model flux…" (page 1383, Section 3.2)*
- *Bushinsky et al. (2019): "For SOSE, the neural network-derived SOCAT-only Southern Ocean uptake was 0.38 Pg C/yr stronger than the true model, while the SOCAT+SOCCOM uptake was 0.26 Pg C/yr stronger than the model…" (page 1384, Section 3.2)*
- *Hauck et al. (2023): "Both mapping methods overestimate the mean $CO_2$ uptake 2009-2018 and the trend 2000-2018 in the SOCAT sampling scheme. In the MPI-SOM-FFN method, the 12% overestimation of the mean in the SOCAT scheme is reduced to 9% in bgcArgo. The 9% overestimation in CarboScope (SOCAT) vanishes in the bgcArgo scheme" (page 9, "Air-sea $CO_2$ fluxes").*

*Both these studies state that adding floats to SOCAT leads to a weaker mean sink. Our study shows the opposite - adding USV observations leads to a stronger mean sink.*

*In the manuscript we emphasize that it is the conclusion of these other studies that ML methods overestimate the $CO_2$ sink (lines 890-802): "**These studies showed** that SOCAT sampling alone overestimates the $CO_2$ uptake in the Southern Ocean, and that additional floats reduce this overestimation, leading to a decreased (weakened) ocean carbon sink."*

pCO2 generally increases over time and mapping methods tend to underestimate pCO2 (thence overestimate fluxes) based on sparse training datasets which have not covered the full range of realistic pCO2 values (many regions with high pCO2 values are unobserved). It's questioning about the distinction between the results in this study and the previous.

*We agree that $pCO_2$ observations are sparse and do not fully cover the distribution of $pCO_2$ space. This is shown nicely by Hauck et al. (2023).*

*Whether the algorithm over- or underestimates $pCO_2$ appears to depend on both the type of reconstruction method used but also the type of testbed used (which models, which member, and whether conclusions are based on an ensemble or models or not). The figure below compares the 'model truth' $pCO_2$ field for the three individual models used in our testbed. As shown on left, the model mean $pCO_2$ fields differ. CanESM2 has higher $pCO_2$ in the Southern Ocean compared to CESM and GFDL. That CanESM2 is higher in the Southern Ocean may be related to its lower*

*bias (right), but more work will be required to confirm such a relationship. At this point, we can see that the choice of model and ensemble member in a testbed matters to reconstruction bias, and thus it is reasonable for us to discuss this potential impact on the comparison of our results to those of Hauck et al. 2023.*

**Model truth pCO₂ field (mean 1982-2016)
for individual models of the LET**

[Figure]

**Ensemble spread of mean bias (over Southern Ocean, 2006-2016)
for 'SOCAT-baseline' for individual models of the LET**

[Figure]

*Hauck et al. (2023) use a 'SOCAT', a 'SOCAT+SOCCOM' and an 'idealized float' sampling mask and one model as the "testbed" (FESOM-REcoM) in their study. They use two different reconstruction methods. As shown by the figure below (their figure 4; see below), the two reconstructions result in different fluxes. The MPI-SOM-FFN method predicts a larger carbon uptake compared to CarboScope. This tells us that the choice of reconstruction method matters.*

[Figure]

[Figure]

Lines 846-854 (in the manuscript with track changes) will need to be revised (or excluded). For such sensitivity tests, one would not expect to see the comparison in performance of different mapping methods but of a fixed method to different sampling scenarios.

*We fully agree with the reviewer. In order to directly compare our study and Hauck's, we would have to use the same sampling mask and testbed model(s), and also calculate the air-sea $CO_2$ flux in the same manner, and then compare the reconstructions using different ML methods. To resolve this, more experiments are needed and these would be beyond the scope of this study. This is the point we aim to convey in this paragraph, to which we have revised to add note of the additional importance of sampling masks (lines 904-907):*

*"Our study and Hauck et al. (2023) use **different sampling masks** and approaches for the calculation of fluxes, which could also be a factor. Targeted, coordinated studies using multiple reconstruction approaches with consistent testbed structures, **sampling masks** and experimental approaches are clearly needed (Rödenbeck et al., 2015)."*

Even in this study, an ensemble of model output or the methods based on SST-removal effects from pCO2 would add more uncertainty to statistics such as bias, RMSD,... That's why I have suggested analyzing further differences in fluxes' variability (trends, seasonal cycles,...) with respect to different sampling strategies.

*As mentioned in our response to Reviewer 1, through comparisons to independent data, Bennington et al. (2022) have demonstrated a marginally improved skill of this reconstruction approach compared to other published observation-based products (this is mentioned in lines 211-212). They also demonstrate lower RMSEs for reconstructions using $pCO_2$-Residual vs. $pCO_2$ (their figure S1), which indicates that the removal of temperature from the target variable enhances the performance of the method.*

*We add note of this in the revised manuscript, with specific mention of Figure S1 of Bennington et al. (2022) (lines 208-210):*

*"Bennington et al. (2022a) demonstrate higher skill for reconstructions using $pCO_2$-Residual as the target variable as opposed to $pCO_2$ (Figure S1 in Bennington et al., 2022a), indicating that the removal of the temperature-driven component enhances the performance of the method."*

[Figure]

*Figure S1 in Bennington et al. (2022), comparing unseen and test RMSE using the Large Ensemble Testbed. ORIG = reconstruction using pCO₂ (no removal of the pCO₂-T component). RESID = reconstruction using the pCO₂-Residual.*

The discontinuity in Figures 3 and 7 still persists: we have obviously seen the gradients in RMSD at (SOCAT or zigzag) sampling tracks versus the "unobserved" areas. Therefore, I expect the authors to verify whether their mapping method put much higher weights on sampled locations than "unobserved" regions ('overfitting': i.e. over-exploitation of the entire available data for model training).

*Figure 3 shows RMSE and bias for the 'SOCAT-only' experiment. As expected, bias and RMSE are higher at times/locations where SOCAT observations are scarce (e.g., in the Southern Ocean). Figure 7 shows the improvement in RMSE when USV observations are combined with SOCAT. RMSE improves mostly in the Southern Ocean where RMSE was initially high. New observations most improve predictions in similar areas because the augmented training set now contains these similar points. This is consistent with autocorrelation lengths for pCO₂ up to 400 km in the Southern Ocean (Jones et al. 2012).*

*Jones, S.D., Le Quere, C. and Rodenbeck, C.: Autocorrelation characteristics of surface ocean pCO2 and air-sea CO2 fluxes. Global Biogeochemical Cycles, 26, 2, https://doi.org/10.1029/2010GB004017, 2012.*

*Following the reviewer's comment, we further analyzed our algorithm to explore overfitting. Indeed, we find some evidence of this, i.e. a statistically significant difference between train and test set error (see figure below). This means that further tuning of the hyperparameters of our ML algorithm could increase generalization skill. But it is important to emphasize that this finding does not invalidate the test or unseen statistics that we present – it simply indicates that more tuning might further improve algorithmic skill.*

[Figure]

*Global mean RMSE (1982-2016) for full, unseen, train and test sets for the 'SOCAT-baseline' experiment. The boxplot shows the ensemble spread of six members of the LET.*

*The goal of this study is to explore how USV sampling added to the Southern Ocean would change skill with all else held equal. We use the same algorithmic approach (same hyperparameters, model is retrained) to reconstruct with only SOCAT or with SOCAT + USV sampling. Further fine-tuning of the algorithm, already shown to perform well by Bennington et al. (2022), is not required to test sampling patterns. Though we don't do the further tuning here, we will take advantage of this useful insight in future work with real-world observations and attempt to further optimize the algorithm to maximize skill.*

*It is worth noting that most ML studies have only the test data with which to estimate generalization skill. Here, we also have unseen data. The plot above shows that test and unseen errors are similar, and thus quoting the test error as a proxy for generalization error appears to be reasonable, if slightly optimistic, for real-world studies where unseen data are not available. More investigation of test and unseen statistics is warranted to better inform real-world uncertainty estimates.*

*We added **Supplementary Text A** and Supplementary **Figure S6** (the figure above), and included the following text in the revised manuscript (lines 417-423):*

*The predicted $pCO_2$ is thus more accurate in areas similar to and surrounding the SOCAT "observations" (i.e., monthly 1°x1° grid cells equivalent to SOCAT coverage, but sampled from the LET). **Figure 3** shows mean bias and RMSE for the full reconstruction (see **Section 2.3**), but note that there is a statistically significant difference between the train and test set errors (**Fig. S6**). This indicates potential overfitting in our ML model (i.e., higher errors for the 'unseen' reconstruction), and that further tuning of the hyperparameters could increase generalization skill (see **Supplementary Text A**).*

**Supplementary Text A:**

*"The hyperparameters for the XGB algorithm used in this study were fixed for all experiments. As we are comparing how sampling impacts the reconstruction, changing the decision trees and depth levels for each experiment would make it difficult to assess whether or not potential changes in bias and RMSE are due to the different sampling strategies or the optimization process. However, **Figure S6** demonstrates a statistically significant difference between train and test set error for the 'SOCAT-baseline' experiment, which may indicate overfitting in our ML model. This suggests that further tuning of the hyperparameters of our ML algorithm might increase generalization skill, and thus reduce test and 'unseen' reconstruction errors.*

*However, further tuning of the algorithm is not the purpose of this study nor is it necessary for the evaluation performed here. As the only factor we change between the experiments is additional Southern Ocean sampling (e.g., the SOCAT mask, algorithmic approach and hyperparameters are the same), we can compare the experiments and understand how different sampling patterns and strategies would change skill in $pCO_2$ reconstructions compared to SOCAT-sampling only.*

***Figure 3** (in main text) shows that errors are higher in locations where SOCAT observations are scarce, such as in the Southern Ocean. The improvements in bias and RMSE when USV observations are combined with SOCAT generally occur at times/locations where errors were originally high, and in the Southern Ocean where the new USV "observations" originate from (e.g., **Figs. 4 and** 7). The additional USV observations thus most improve*

*predictions in surrounding areas because the augmented training set contains these similar points. However, note that the lower error values shown for the training set do not impact the final error metrics presented in our study, as ~ 99 % of the reconstruction consists of 'unseen' data points (Figs. S1, S2)."*

From a statistical point of view, different mapping methods learned on different model testbeds (i.e. different training data have different data ranges) probably result in different magnitudes of RMSE or Bias. It is not convincing to mention that their mapping method has error values in line with those in the previous study.

*We agree that this comparison is not precise. We wanted to show that our error values were not significantly different compared to related studies.*

Again, in the following sentence and others in the text, please be careful using the phrase "Observation-based data products". Precisely, "mapping methods" have been developed to estimate pCO2 and generate global "Observation-based data products". Lines 50-52 (in the manuscript with track changes): "Observation-based data products have been developed to estimate full-coverage surface ocean pCO2 across space and time by extrapolating to global coverage from these sparse *SOCAT observations.*"

*We have replaced "observation-based data products" with "mapping methods" (lines 50 and 212).*